# Microbial and Biogeochemical Shifts in a Highly Anthropogenically Impacted Estuary ("El Sauce" Valparaíso)

**Francisco Pozo-Solar** [1,2,3], **Marcela Cornejo-D'Ottone** [4], **Roberto Orellana** [2,3], **Carla Acuña** [3], **Cecilia Rivera** [2,3], **Polette Aguilar-Muñoz** [3], **Céline Lavergne** [3,5] and **Verónica Molina** [2,3,6,*]

[1] Programa de Doctorado Interdisciplinario en Ciencias Ambientales, Facultad de Ciencias Naturales y Exactas, Universidad de Playa Ancha, Valparaíso 2340000, Chile; panchopozosolar@gmail.com

[2] Departamento de Ciencias y Geografía, Universidad de Playa Ancha, Avenida Leopoldo Carvallo 270, Playa Ancha, Valparaíso 2340000, Chile

[3] HUB Ambiental UPLA, Universidad de Playa Ancha, Leopoldo Carvallo 207, Playa Ancha, Valparaíso 2340000, Chile; celine.lavergne@upla.cl (C.L.)

[4] Escuela de Ciencias del Mar and Instituto Milenio de Oceanografía, Pontificia Universidad Católica de Valparaíso, Valparaíso 2950, Chile

[5] Laboratory of Aquatic Environmental Research, Centro de Estudios Avanzados, Universidad de Playa Ancha, Viña del Mar 450, Chile

[6] Centro de Investigación Oceanográfica COPAS COASTAL, Universidad de Concepción, Concepción 4070386, Chile

\* Correspondence: veronica.molina@upla.cl

**Abstract:** Coastal zones are ecosystems that are sensitive to climate change and anthropogenic pollution, resulting in a potential loss of biodiversity and ecosystem services through eutrophication and nutrient imbalances, among others. The coastal El Sauce catchment area, Central Chile, is under multiple anthropogenic pressures including wastewater treatment plant (WWTP) discharge, which its broad effect remains underexplored. In order to assess the impact of the WWTP on El Sauce stream, the benthic microbial communities and key functional groups variability (i.e., nitrifiers, methanogens and methanotrophs) were determined by 16S rDNA high-throughput sequencing and by functional genes quantification, respectively, during two contrasted seasons in three catchment areas (pre-, WWTP and post-discharge). The microbial communities' structure profiles were associated with the water quality, nutrients, greenhouse gas (GHG) distribution, and the organic matter isotopic signatures in the sediments, for the first time, in this ecosystem. The results show that organic matter isotopic signatures using nitrogen and carbon ($\delta^{15}N$ and $\delta^{13}C$) and the physicochemical conditions in El Sauce estuary changed from the pre- to WWTP discharge areas (i.e., a pH decrease of 0.5 units and an increase of 4–6 °C in the water temperature). The WWTP discharge area was characterized by a low nutrient concentration and significantly higher GHG distribution (>600 μM $CO_2$, >30,000 nM $CH_4$, and >3000 nM $N_2O$). In addition, the benthic microbial community structure shifted spatially and seasonally, including specific phyla known as sewage bioindicators, such as Firmicutes (Clostridiales order) and Bacteroidetes. In addition, other taxa were enriched or only retrieved in the sediments of the WWTP influenced area, e.g., Tenericutes, Lentisphaerae, Synergistetes, and LCP-89. Methanogens were more enriched near the WWTP discharge compared to those in the pre-discharge site in both seasons, while methanotrophs and ammonia oxidizers were unfavored only during winter. Our results indicate that the WWTP discharge impacts the biogeochemical conditions in El Sauce catchment area modifying the benthic microbial communities, including a decrease in the key functional groups able to mitigate $CH_4$ and regulate nutrients recycling in these aquatic ecosystems.

**Keywords:** wetland; greenhouse gases; organic matter; benthic microbial community; nitrifiers; methanotrophs; methanogens

## 1. Introduction

Organic matter (OM) is a central factor regulating nutrient recycling in several aquatic ecosystems. The processes involved in controlling the transformation of OM are especially relevant in coastal environments, such as estuaries and wetlands, since they contribute to global carbon mitigation [1]. Estuarine OM is composed of a diverse mixture of allochthonous and autochthonous sources [2]. While autochthonous OM enriched by plant-derived polymers is locally originated from the high primary productivity associated with phytoplankton and aquatic vegetation, allochthonous OM is often advected from rivers, tributaries, groundwater, or allocated sources mainly from a terrestrial origin. The seasonal distribution of OM sources, its quantity, quality, and environmental physical and chemical characteristics are key factors governing the biogeochemical processes in estuaries. Therefore, it affects the net flux of nutrients to coastal marine ecosystems. Several of those elements are currently threatened by climate change and anthropogenic activities [3–5]. Indeed, many estuaries receive industrial, domestic, agricultural, and wastewater discharges, which often are highly enriched by fractions of sewage and fossil fuels, and may include complex OM mixtures [6,7] and other pollutants [8,9] resulting in eutrophication and biodiversity loss [10] and affecting aquatic ecosystem functions [11].

Among the other environmental factors, the origin, composition, and distribution of OM play a relevant role in reshaping the structure and composition of natural microbial communities of estuaries [12–14]. Anthropogenic OM sources and compositions were reported to influence water quality and microbial community taxonomic composition and structure in aquatic ecosystems [15]. Wastewater treatment plants (WWTP) are facilities in which several processes aim to remove solids, as well as transform liquid wastes into an acceptable final effluent. The partial degradation of OM and nutrients mineralization occur through a series of stages, such as activated sludge, aeration tanks, bioreactors, and disinfection processes before water is discharged to the environment. However, multiple studies have reported that WWTPs are important sources of several contaminants that may negatively impact aquatic ecosystems, including those of emerging concern [16–18]. Besides persistent chemical residuals, OM, and microorganisms remain in treated waters influencing both the bio-physical and ecological conditions, especially when the receiving water body is a semi-closed aquatic transition ecosystem, such as estuaries with higher water residence times [19–22]. Several lines of evidence suggest that wastewater discharge in estuaries increases the amount of greenhouse gases (GHG) emitted into the atmosphere [16,23,24], as well as other freshwater ecosystems, such as river mouths [25] and lakes [26]. WWTP discharges carry over a significant proportion of remaining microbes associated with the human gut microbiome, including pathogens and other microbial communities besides Coliform bacteria. These microbes are typical indicators of wastewater that can shape the natural community structure and substantially affect their biogeochemical activity [27]. In addition, WWTP discharge changes the water quality by increasing its temperature, altering the nutrient concentration and stoichiometry, as well as, adding antibiotics and other emerging pollutants that affects the microbial community structure [28]. Moreover, WWTP could cause shifts in the functional microbial groups, for instance, methanogenic communities are favored, while methanotrophs and phototrophs are unfavored in WWTP-impacted zones [29,30].

El Sauce catchment, located in Laguna Verde, Valparaíso, has been considered as a eutrophic system for several years mainly due to runoff and loading of various sources of pollution from the constant water discharge of a WWTP, as well as inputs from leachates from a municipal landfill and domestic wastewater [31]. Nevertheless, it is unknown how these disturbances influence GHG and the native microbiome including functional groups associated with GHG recycling. This study aimed to determine the influence of WWTP discharge on the overall biogeochemical conditions along El Sauce estuary in winter and summer, including the changes in the dissolved nutrient and GHG concentration ($CO_2$, $CH_4$, and $N_2O$), OM quality, and benthic microbial community structure.

## 2. Materials and Methods

### 2.1. Study Area and Sampling Procedures

El Sauce estuary is located in Valparaíso (Central Chile; 33°6′ S—71°38′ O), and its main water flow from Las Cenizas basin area is connected with "La Luz" reservoir. The studied stream is situated within the Peñuelas sub-watershed, representing a total of 320.9 km$^2$ (DGA, 2014). [32] As a part of the coastal watershed complex called "Aconcagua/Maipo Coast", the water flow is mainly dependent on rainfall and anthropogenic activities. It is important to note that a sandbar avoiding superficial water exchanges between the stream and the ocean is almost constantly present at the estuary and was present during both sampling periods. Field work to collect physicochemical data water and sediments samples was conducted on August 26, 2019 (austral winter) and January 07, 2020 (austral summer). Five sites were sampled along the main tributary and estuary mouth of the Las Cenizas basin following a previous report (Rivera-Castro et al., 2020) [31]. Two sampling sites from the upper area of the main tributary were located 30 m upstream (F2) and 30 m downstream of the WWTP discharge (F3). A third site (E7) was located at an intermediate tributary area characterized by a pond. The last two sampling sites were situated near the estuary (E10) and in the mouth (E11) (Figure 1). During winter and summer, the tributary upper and intermediate portions were shallow, with a 10–20 cm water depth, whereas at the estuary mouth the water depth was approximately 50–60 cm.

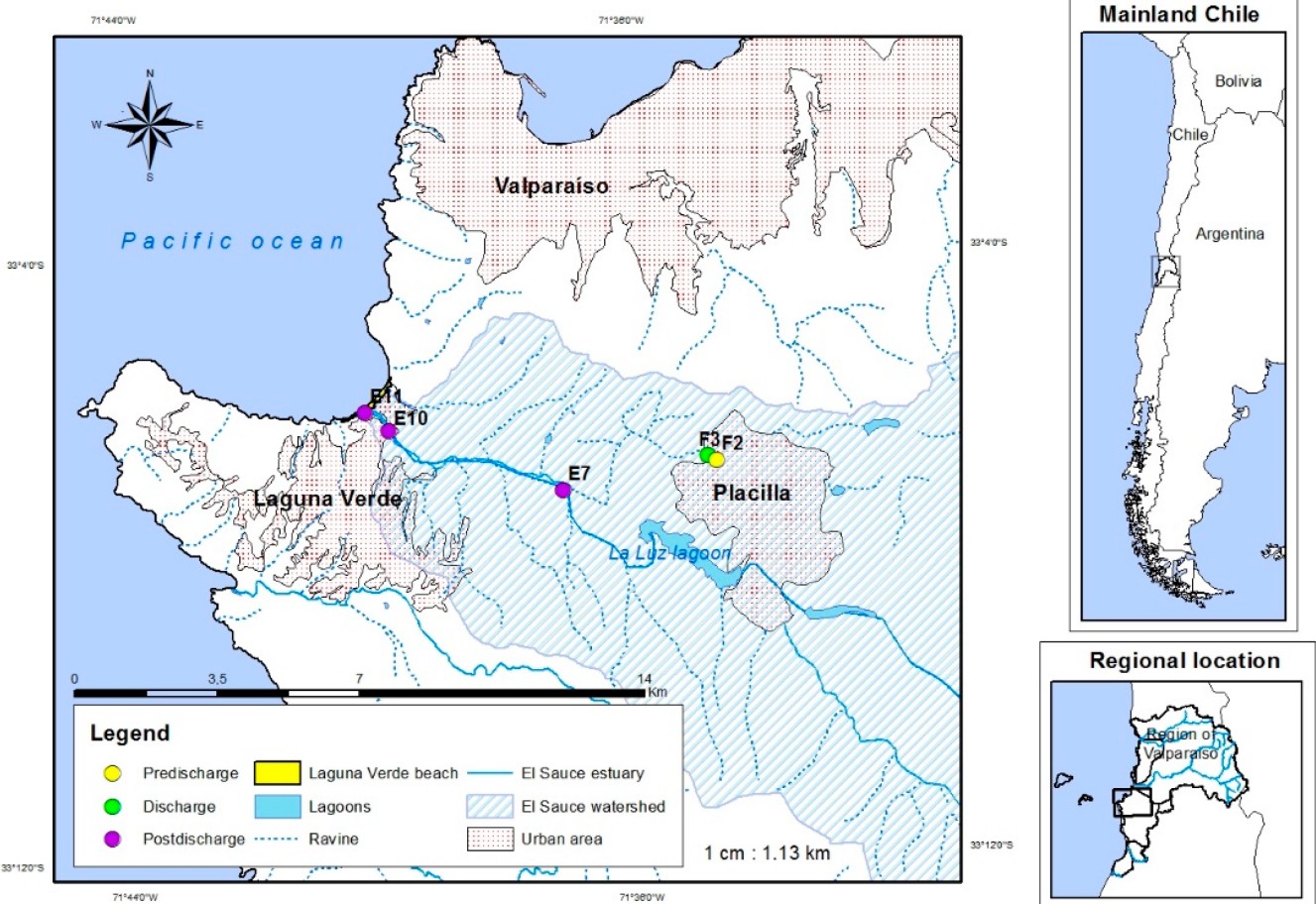

**Figure 1.** Sampled sites of El Sauce estuary including five stations.

Physicochemical parameters such as temperature, pH, dissolved oxygen (DO), and conductivity were recorded in the surface water in triplicate (except for DO) using a multiparameter probe (YSI and ProDSS model). Triplicate water samples were collected from the water surface for nutrients, including nitrate ($NO_3^-$), nitrite ($NO_2^-$), phosphate ($PO_4^{3-}$), and silicic acid ($SiO_4^{4-}$) determinations. The water was filtered through a 0.45 μm GFF filter and collected into 60 mL "Nalgene" bottles (Nalgene, Waltham, MA, USA). These bottles were kept cool for less than 3 h and frozen at −20 °C until further analysis. Additionally, water samples were collected to determine the concentration of dissolved GHG ($CO_2$, $N_2O$, and $CH_4$) using 20 mL gas-tight vials, in triplicate. These vials were filled under water to avoid bubbles, then, were immediately fixed with 50 μL saturated $HgCl_2$ solution, and stored at room temperature in the dark until their analysis in the laboratory. Water samples that met the standard water quality parameters, according to Chilean regulation NCh409/84, were also collected in triplicate and transported to the laboratory following procedures established by the Standard Methods Book [33]. For the determination of Chlorophyll-a concentration, in triplicate, discrete water samples (500 mL) were filtered onto GF/F filters (Whatman® glass microfiber filters, USA, Sigma Aldrich), and then stored and frozen until laboratory analyses.

Surface sediment samples were sampled in sealable bags for organic matter and granulometry determinations and were kept at 4 °C for further laboratory analyses. Fine-grained sediment samples were collected to determine carbon ($\delta^{13}C$) and nitrogen ($\delta^{15}N$) isotopes and stored in cryotubes (2 mL, in duplicate). Additionally, fine-grained sediments were sampled for molecular microbial community composition analysis in cryotubes (2 mL, in duplicate) filled with RNA later (Ambion, Life Technologies, Carlsbad, CA, USA, Thermo Fisher Scientific) to preserve nucleic acids [34]. All the sediment samples were transported under cool conditions using gel packs (less than 3 h). Sediments for isotopic and microbial community analyses were stored frozen at −80 °C until analyses.

## 2.2. Laboratory Procedures for Biochemical Parameters

Nutrients concentrations were determined through spectrophotometric methods. Nitrate and nitrite concentration were determined following Strickland et al. (1972) [35] and $PO_4^{3-}$ and $SiO_4^{4-}$ were measured following Atlas et al. (1971) [36] using a nutrient autoanalyzer. Chlorophyll-a concentration was determined in discrete seawater samples according to the method described by Holm-Hansen et al. (1965) [37]. Dissolved GHG concentrations in water were determined by gas chromatography through the headspace technique McAuliffe (1971) [38] using a Greenhouse GC-2014 Shimadzu chromatograph equipped with an electron capture detector (ECD) for $N_2O$ and a flame ionization detector (FID) for $CO_2$ and $CH_4$ attached to a methanizer for the conversion of $CO_2$ into $CH_4$. Calibration was carried out using three calibration points using helium, air, and a standard ratio of 600 to 5, to 1 of a $CO_2$, $CH_4$, and $N_2O$ mixture for absolute quantification (Scotty gas mixture; Air Liquid Co., Paris, France). Standard water quality parameters such as transparency, turbidity, color, dissolved total solids (DTS), sulfates, chlorides, fecal coliforms, and $BOD_5$ (biochemical oxygen demand) according to Chilean regulation NCh1333/78 were measured following the Standard Methods Book [33]. All the chemicals used for the above analyses were of analytical grade.

## 2.3. Sediment Granulometry, Organic Matter, and Isotopic Analyses

Sediment granulometry of samples were assessed using dry sediment (40 °C during 24 h) and analyzed by standard mesh analysis method to sort out the sand, gravel, and silt size fractions (Friedman and Sanders 1978) [39]. Total organic matter was determined using the Loss-On-Ignition (LOI) method in a muffle furnace [40]. Carbon ($\delta^{13}C$) and nitrogen ($\delta^{15}N$) isotopes were analyzed using mass spectrometry (Thermo Scientific Delta V Advantage IRMS using an EA-2000 Flash Elemental analyzer, Waltham, MA, USA) according to Pee Dee Belemnite (VPDB) and the air standards of the Laboratory of Biogeochemistry and Applied Stable Isotopes (LABASI, PUC, Chile).

### 2.4. Benthic Microbial Community Characterization Using Molecular Approaches

DNA was extracted using the DNAeasy PowerSoil DNA Isolation Kit (Qiagen, Germantown, MD, USA). Briefly, after thawing, the RNA later buffer was removed from the sample, then 250 mg of wet sediment was weighted and distributed into the PowerBead tubes provided by the manufacturer. DNA extracts were quantified (1.3–40 ng/μL) using the dsDNA BR (Broad Range) Qubit 2.0 fluorometer (Thermo Fisher Scientific, Waltham, MA, USA), and DNA quality (260/280) was determined by spectrophotometry using Cytation 5 (BioTek, Shoreline, WA, USA), and template amplification was tested using conventional 16S rDNA PCR using 27F and 519R primers. DNA extracts (*n* = 10) were sent to the Illumina Miseq sequencing platform in Mr. DNA laboratory (Shallowater, TX, USA) to be sequenced using 515F (GTGYCAGCMGCCGCGGTAA) [41] and 806R (GGACTACNVGGGTWTCTAAT) primers [42] for the V4 region of the 16S rDNA gene. The sequences were deposited in the European Nucleotide Archive (ENA) under project accession ID PRJEB44347 and primary accession samples (ERR9814522–ERR9814533). The stations were separated into three categories, e.g., pre-discharge (F2), discharge (F3), and post-discharge (E10 and E11). Samples from station E7 were not included in the microbial communities' analysis. The same criteria were used in the qPCR analyses.

### 2.5. Quantitative PCR Analyses

Ammonia oxidizing bacteria from the Betaproteobacteria (βAOB), comammox *Nitrospira* sp, Methanotrophic bacteria associated with *Methylobacter*-like ones, and Methanogenic archaea functional groups were quantified using the following functional genes *amoA, comaA, pmoA,* and *mcrA*, respectively, with the primer references and qPCR conditions shown in Table 1. The qPCR reactions used approximately 10 ng of DNA as a template and 20 μL reactions using the Power SYBR Green Master Mix (Applied Biosystems, Waltham, MA, USA) using the QuantStudio 3 thermocycler (Applied Biosystems, Waltham, MA, USA).

**Table 1.** qPCR table indicating its gene, group, primers, primers concentration, annealing temperature, efficiency, $R^2$, and reference.

| Gene | Group | Primers | Primer Concentration (μM) | Annealing Temperature (°C) | Efficiency (%) | $R^2$ | Reference |
|---|---|---|---|---|---|---|---|
| **Ammonia monooxygenase subunit A (*amoA*)** | Betaproteobacteria βAOB | amoA1F amoA2R | 0.4 0.4 | 56 | 92.55% | 0.998 | [43] |
| **Ammonia monooxygenase subunit A (*comaA*)** | COMAMMOX Nitrospira Clade A | comaA-244F coma-659R | 0.4 0.4 | 52 | 83.10% | 0.999 | [44] |
| **Methane monooxygenase subunit A (*pmoA*)** | Methanotrophs | A189F mb661R | 0.4 0.4 | 55 | 90.34% | 0.999 | [45] [46] |
| **Methyl coenzyme M reductase subunit A (*mcrA*)** | Methanogens | mlasmcrA-rev | 1 1 | 55 | 90.59% | 0.999 | [47] |

Data were collected and further analyzed by the software QuantStudioTM Design and Analysis Software v1.5.2. The reactions were run following the manufacturer's recommendations and previous protocols available in the laboratory for βAOB [48] and methanogens [49]. All the reactions were run in triplicate using the standards available in our laboratory (Table 1), and the results are expressed as gene copies per gram of wet sediment. Detection limits were confirmed by the dynamic range of the curves, which were above CT 31–33 (~40 copies × μL), and the expected amplicon peak during the melting curve inspection.

### 2.6. Sequencing Data Curation and Taxonomic Classification Processing

QIIME2 (v.2019.7) was used for sequence curation and taxonomic classification into Amplicon Sequence Variant (ASVs) [50]. First, the sequences were imported to QIIME2

using Casava. Then, the sequences were demultiplexed, and poor-quality or short sequences (<210 bp length) were removed using qiime dada2 denoised-paired plugin [51]. The summer samples from E7 contained a few sequences, and thus, were removed for further analyses. Taxonomic classification was performed using the SILVA132 database (V4 subunit of the 16S rRNA gene region) with the feature-classifier classify-consensus-vsearch plugin [52].

### 2.7. Statistical Analyses

All the statistical analyses were carried out using R software [53]. Because the normality assumption cannot be verified with our data, Spearman correlations were conducted to evaluate the relationships between the physical and chemical conditions in water and sediments using the rcorr function with the 'Hmisc' package [54]. Graphical visualization, including significance, were generated using the corrplot package [55]. Taxa bar plots for phylum and functional groups related to GHG recycling spatial and winter/summer changes were generated using the 'phyloseq' package [56]. In addition, to determine the potential changes associated with the influence of WWTP discharge in the benthic microbial community structure and their association with environmental conditions, the samples were grouped as: pre-discharge (F2), discharge (F3), and post-discharge stations (E10 and E11) and plotted using the amp_heatmap function of the 'ampvis2' package [57]. A redundancy analysis (RDA) was performed using the ordinate function of the 'phyloseq' package to determine the relationship between the microbial community (Phylum level) shifts and the environmental factors using the envfit function of vegan package [58] to identify the significance of the correlation. A differential analysis of ASVs was performed using the 'DESeq2' package, specifically, the DESeq function. The data used for the comparison are the ASV of each crude sample, without normalization. The results are visualized in a Volcano Plot [59]. SIMPER (Similarity Percentage) analysis was conducted using PRIMER6 software [60] to identify the phylum contributions between the pre-discharge, discharge, and post-discharge stations.

## 3. Results

### 3.1. Environmental Factors, Nutrients, and Water Quality Variables

The surface water temperature of El Sauce estuary was characterized by lower values in winter (12.1–17.6 °C) compared to those in summer (17.63–26.1 °C), and the maxima were found associated with the WWTP discharge station F3 (Figure 2A,B). The water pH was lower at the upper stations, particularly in F3 station, which was characterized by pH values < 6.86 in winter and summer, compared to those of the post-discharge stations, where pH > 7.4 was detected (Figure 2A,B). Water conductivity was found to increase from the upper portion towards the mouth, ranging from 766 to 1304 $\mu$S cm$^{-1}$ and 828 to 4150 $\mu$S cm$^{-1}$ in winter and summer, respectively. In general, the DO demonstrated spatial conductivity shifts, but with a higher variability in winter compared to that in summer (Figure 2C,D).

The nutrients (i.e., nitrite, nitrate, silicate, and phosphate) at the F2 and F3 stations were characterized by extreme changes in nitrate and nitrite, which reached the lowest values at the WWTP discharge station (F3 station), <14 $\mu$M for nitrate and <3 $\mu$M for nitrite in winter and summer, respectively. The highest values of nitrate and nitrite concentrations were registered in the upper tributary and intermediate pond stations (F2 and E7,) with >200 and $\mu$M > 100 $\mu$M values in summer (Figure 2E,F), respectively. Estuary stations E10 and E11 were characterized by similar nitrate and nitrite concentrations, and slightly higher concentrations towards the mouth (from E10 to E11) during summer (Figure 2E,F). The phosphate concentrations were higher in the upper tributary station (F2), and unlike other nutrients, greater concentrations were found in winter (>15 $\mu$M) than those in summer (<5 $\mu$M). The F3 and E11 stations exhibited lower concentrations (<1 $\mu$M) in both seasons (Figure 2G,H). Except for the upper tributary (Station F2) in summer, silicate presented

lower concentrations during winter (320–100 μM) versus those in summer (100–250 μM) from F3 station to the estuary mouth (Figure 2G,H).

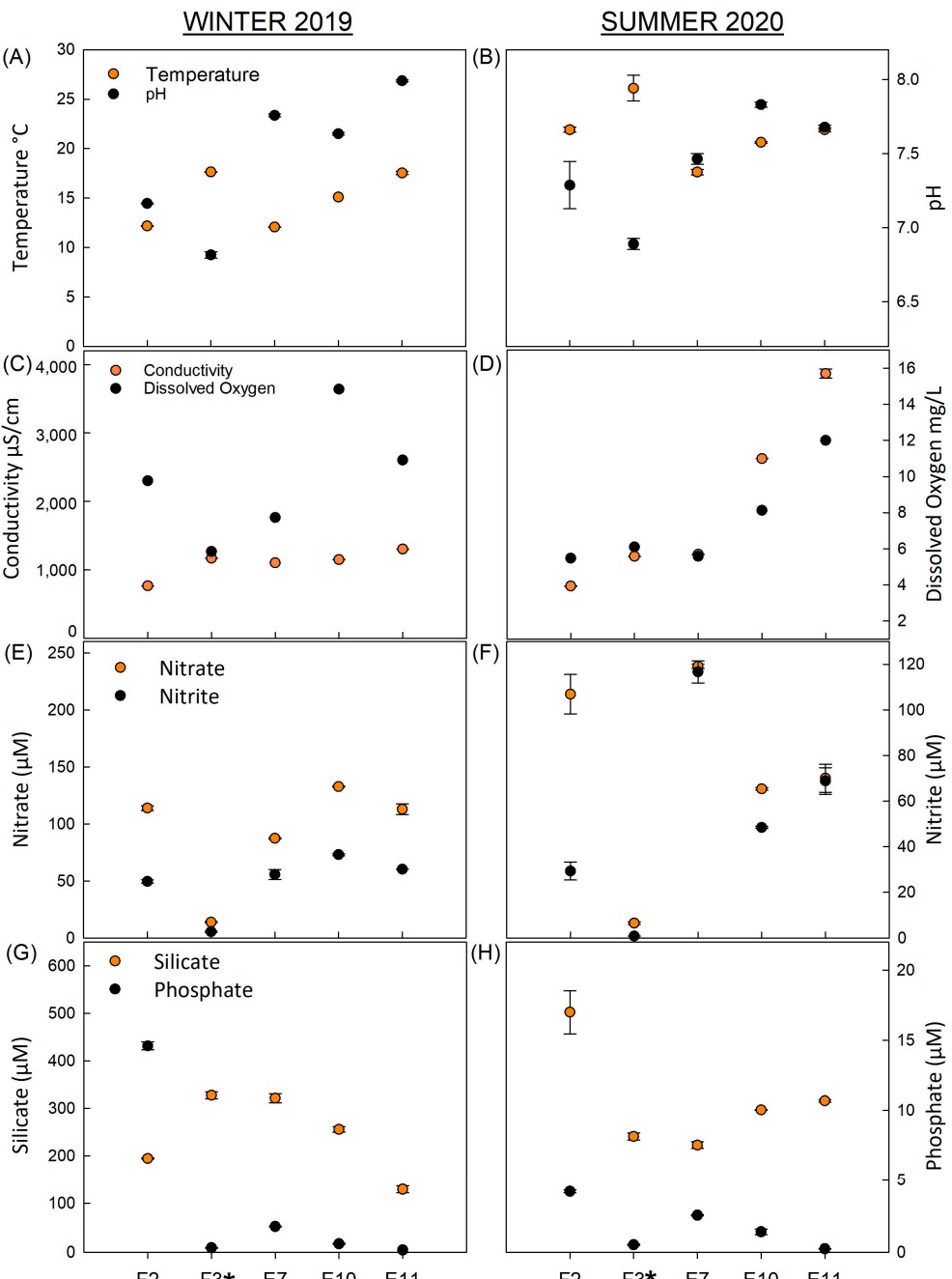

**Figure 2.** Physicochemical and nutrients conditions in all water samples during winter (**left panels**) and summer (**right panel**) of El Sauce estuary; Temperature (**A,B**); Conductivity (**C,D**); Nitrate and Nitrite (**E,F**), Silicate (**G,H**). * Discharge Station.

The water quality conditions based on additional physicochemical variables, such as transparency, color, and dissolved total solid (DTS), exceeded the maximum permitted values for agricultural irrigation and animal consumption according to the environmental Chilean regulation NCh1333/78 (Table 2). Moreover, fecal coliforms also exceeded the allowed values (>1000 CFU/mL) in the upper tributary (F2) and in the mouth (E11) in both the summer and winter seasons.

**Table 2.** Water quality factors in all sampled stations during winter and summer. All bold numbers are values that exceed the maximum allowed limit (MAL). The values in brackets are the results previously reported by Rivera et al. (2020).

| FACTOR | WINTER | | | | | SUMMER | | | | | MAL |
|---|---|---|---|---|---|---|---|---|---|---|---|
| | F2 | F3 | E7 | E10 | E11 | F2 | F3 | E7 | E10 | E11 | |
| DTS (mg/L) | 381 | **587** | **543** | **563** | **647** | 413 | **645** | **660** | **1406** | **>2000** | <500 |
| Transparency (Secchi disk depth in cm) | **18** | **40** | **12** | **60** | **84** | ND | ND | ND | **60** (0.61; 0.2) | **4** (0.11; 0.12) | ≥120 |
| Turbidity (NTU) | 5.54 | 18.53 | 15.60 | 10.02 | 14.27 | 1.93 | 32.83 | **55.40** | 4.91 | 10.16 | 50 |
| Color (Pt/C) | **139.00** | **316.00** | **299.67** | **183.00** | **278.67** | **254.00** (139; 126) | **624.00** (448; 1063) | **715.00** (550; 1572) | **239.00** (415; 75) | **218.00** (350; 225) | <100 |
| Sulfate (mg/L) | 85.33 | 244.00 | 141.33 | 133.33 | 136.00 | 124.85 | 159.15 | 153.66 | 205.80 | **270.28** | 250 |
| Chlorides (mg/L) | 75.63 | 146.53 | 146.53 | 184.34 | **255.24** | 0.70 (70.9; 113.5) | 1.43 (106. 4; 156) | 1.57 (**355; 269**) | 5.30 (142; **326**) | 8.73 (**709; 2822**) | 200 |
| Total Coliforms (NMP) | **9200** | <1.8 | <1.8 | ND | **1100** | **≥16,000** (**>16,000**; **>1600**) | 0 (**>16,000**; <1.8) | 110 | 500 | **2200** | 1000 |
| Fecal Coliforms (NMP) | 140 | <1.8 | <1.8 | ND | 490 | 260 (<1.8; 20) | 0 (20; <1.8) | 80 | 300 | 140 | 1000 |
| BOD$_5$ (mg/L) | 7.30 | 0.00 | 4.35 | 4.51 | 11.98 | 11.3 (**1000**; 5.44) | 65.79 (**1400**; 99.9) | 66.41 | 17.71 | 20.31 | 20 |
| Chlorophyll-a (mg/L) | 6.52 | 0.6 | 0.19 | 20.02 | 41.14 | 31.81 | 0.39 | 39.28 | 22.74 | 32.92 | 10 |

### 3.2. GHG Distribution along El Sauce Estuary

WWTP discharge has a high impact on the dissolved concentration of GHG in the Laguna Verde estuary, which presented a maximum value at station F3 (WWTP discharge site), reaching up to 1200 µM $CO_2$ in summer, and 55,301 nM $CH_4$ and 3930 nM $N_2O$ in winter (Figure 3). These dissolved $CH_4$ concentration values were up to 60 fold higher than they were in the upper tributary pre-discharge water (Station F2). While the WWTP discharge increases the concentration of methane to a greater extent than the concentrations of $N_2O$ and $CO_2$, those gases registered a continuous decrease from the intermediate pond (Station E7) to the estuary mouth (Figure 3).

### 3.3. Sedimentary Conditions and Potential Organic Matter Quality

Sediments were characterized by the predominance of larger particle sizes (i.e., medium–coarse and very coarse ones) in the upper tributary, whereas lower particle sizes (i.e., silt and very fine sand) were found in the sediments of the mouth, especially during winter compared to those of the summer samples (Figure 4). The total organic matter (TOM) increased by >100 % (<4 to >8 g) from the upper tributary (F2 and F3) and intermediate pond (E7) post-discharge WWTP to the mouth stations (E10 and E11) (Figure 4).

The isotopic organic matter signatures coincided with TOM shifts characterized by a marked difference in $\delta^{15}N$ versus $\delta^{13}C$ isotopes ratios, predominantly ~29 $\delta^{13}C$ and ~6 $\delta^{15}N$ in the upper tributary compared to the mouth estuary sediment, ~26 $\delta^{13}C$ and ~12 $\delta^{15}N$ (Figure 5A), showing higher $\delta^{15}N$ ratios in summer compared to those in winter. In addition, higher C/N ratios were found in the WWTP discharge during summer and the intermediate pond water during winter (C/N > 10), whereas lower ratios were determined around the mouth station (E10 and E11) and the intermediate pond water during summer, reaching C/N < 8 (Figure 5B).

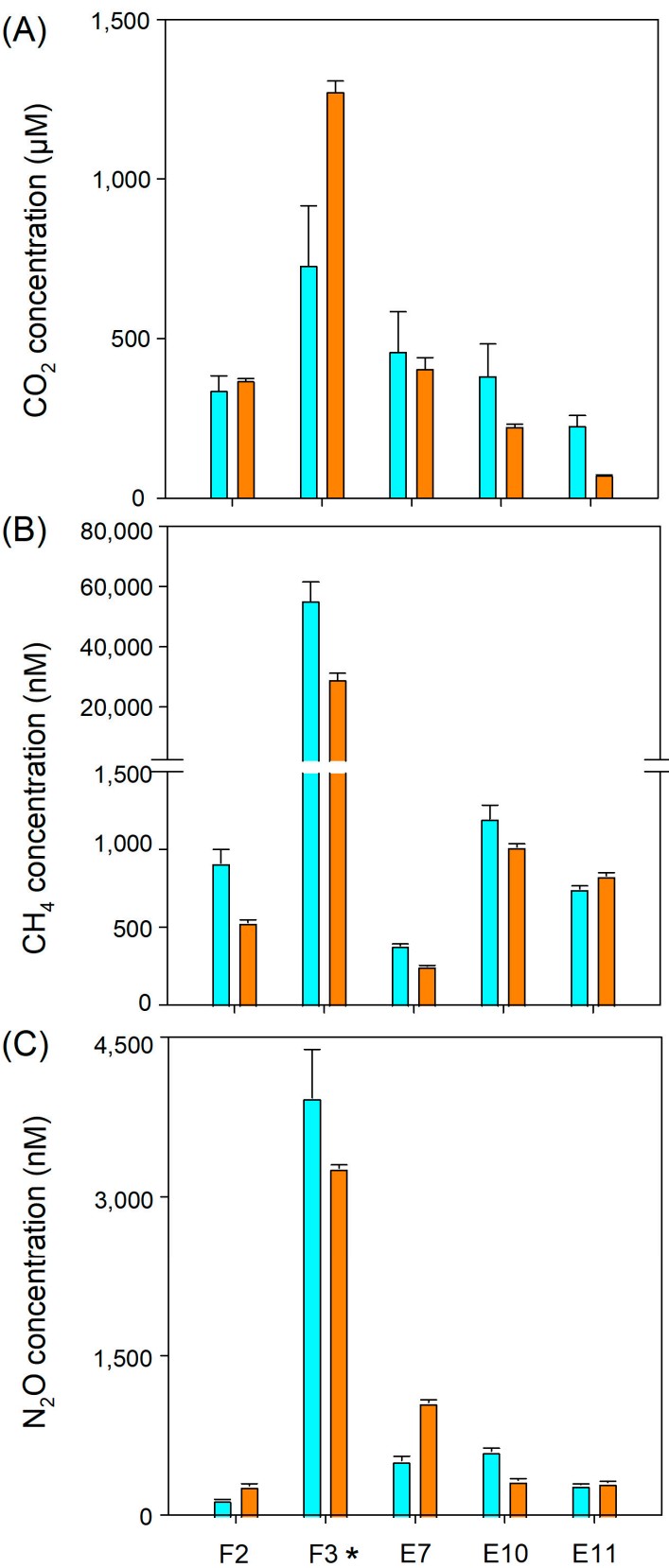

**Figure 3.** Greenhouse gas concentration (**A**) $CO_2$, (**B**) $CH_4$, and (**C**) $N_2O$ dissolved in water in winter (cyan bars) and summer (orange bars) of El Sauce Estuary. * Discharge Station.

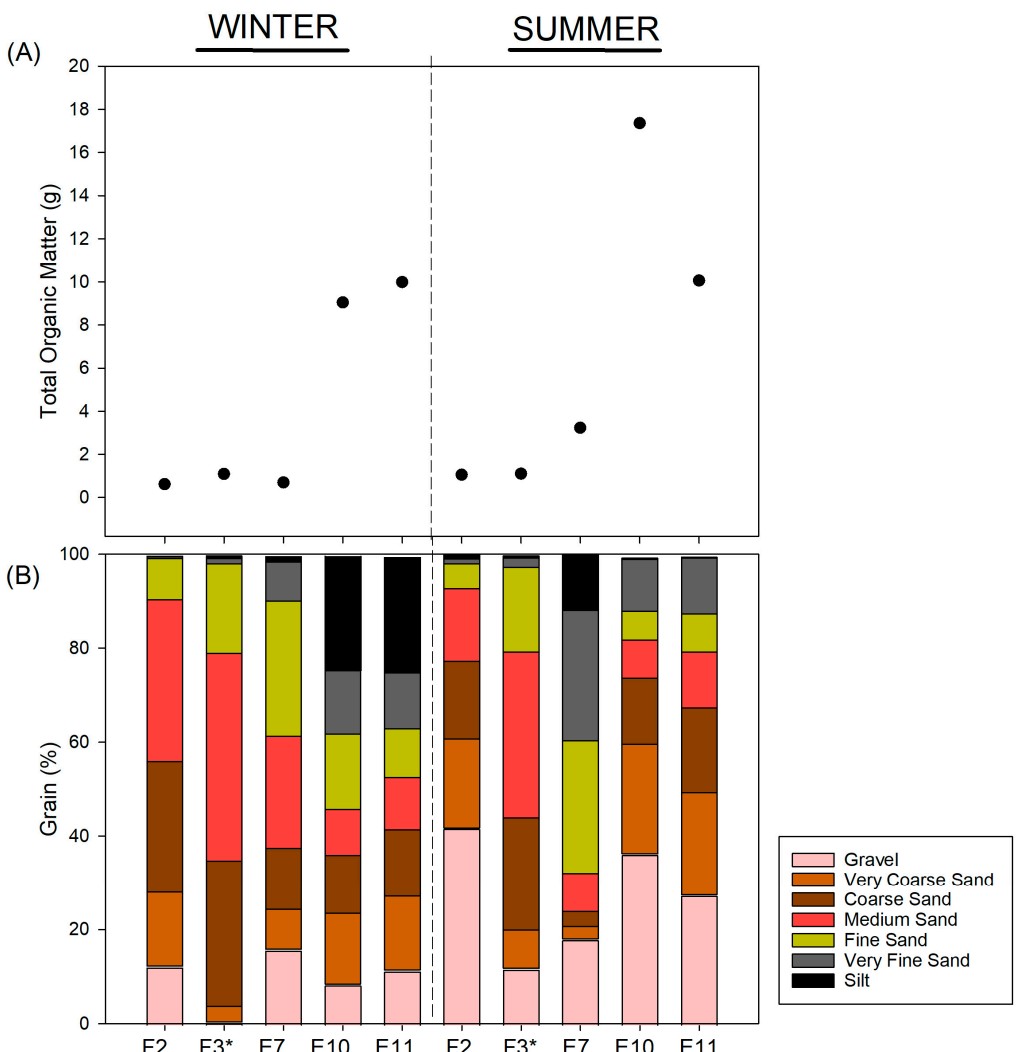

**Figure 4.** (**A**) Total organic matter percentage and (**B**) sediment granulometry distribution. * Discharge Station.

### 3.4. Spearman Correlation Analyses Associated with Water Quality Parameters and GHG

Among the physicochemical parameters, pH and conductivity were significantly linked to the water quality parameters, GHG concentrations, nutrients, and sediments properties (Figure 6). For instance, pH was significantly negatively correlated with GHG in the water and granulometry in the sediments ($p$-value < 0.05), while it was positively correlated with TOM in the sediments ($p$-value < 0.05) (Figure 6). Conductivity was significantly positively correlated with the TDS, sulfates, and sedimentary TOM ($p$-value < 0.05). Water quality variables, such as turbidity, were significantly correlated with color ($p$-value < 0.05) and with both $BOD_5$, TDS, and sulfates ($p$-value < 0.05) in the water and with TOM ($p$-value < 0.05) in the sediments. In addition, Chlorophyll-a was significantly correlated with fecal coliforms, $CO_2$, nitrate, nitrite, and $\delta^{15}N$ ($p$-value < 0.05). Interestingly, GHG was significantly correlated among them ($p$-value < 0.05), and also, with nitrate ($p$-value < 0.05) (Figure 6). Regarding sediment granulometry, medium sand was significantly correlated ($p$-value < 0.05) with those in the different categories and with other sediment variables (Figure 6), for example, with TOM, $\delta^{15}N$ and several water physico-chemical variables, including Chlorophyll-a, GHG, nitrite, and nitrate. The coarse sand percentage was also significantly correlated with pH, Chlorophyll-a, $CH_4$, nitrite, and nitrate ($p$-value < 0.05) (Figure 6).

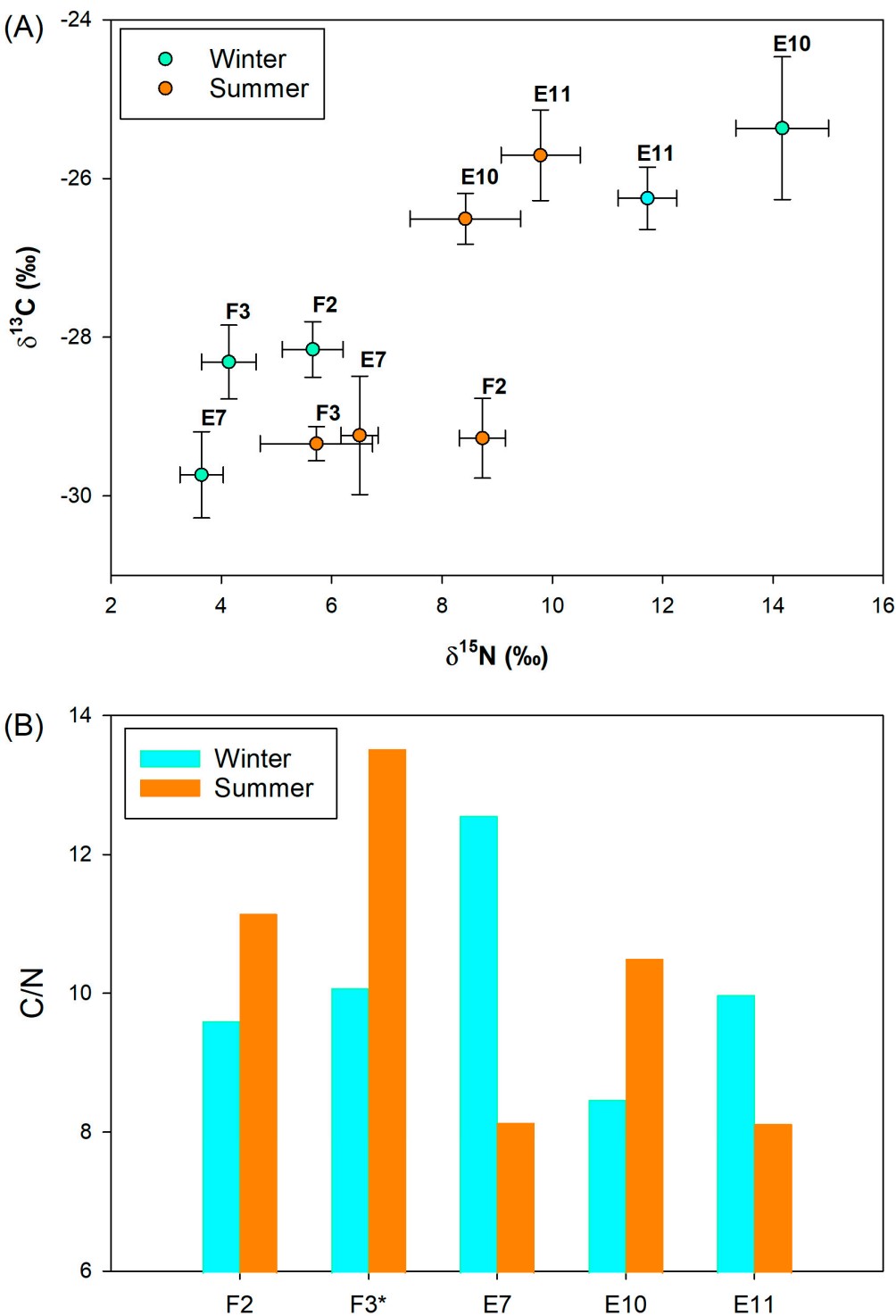

**Figure 5.** Isotopic organic matter signatures (**A**) and C/N rates (**B**) in sediments during winter and summer of El Sauce estuary. * Discharge Station.

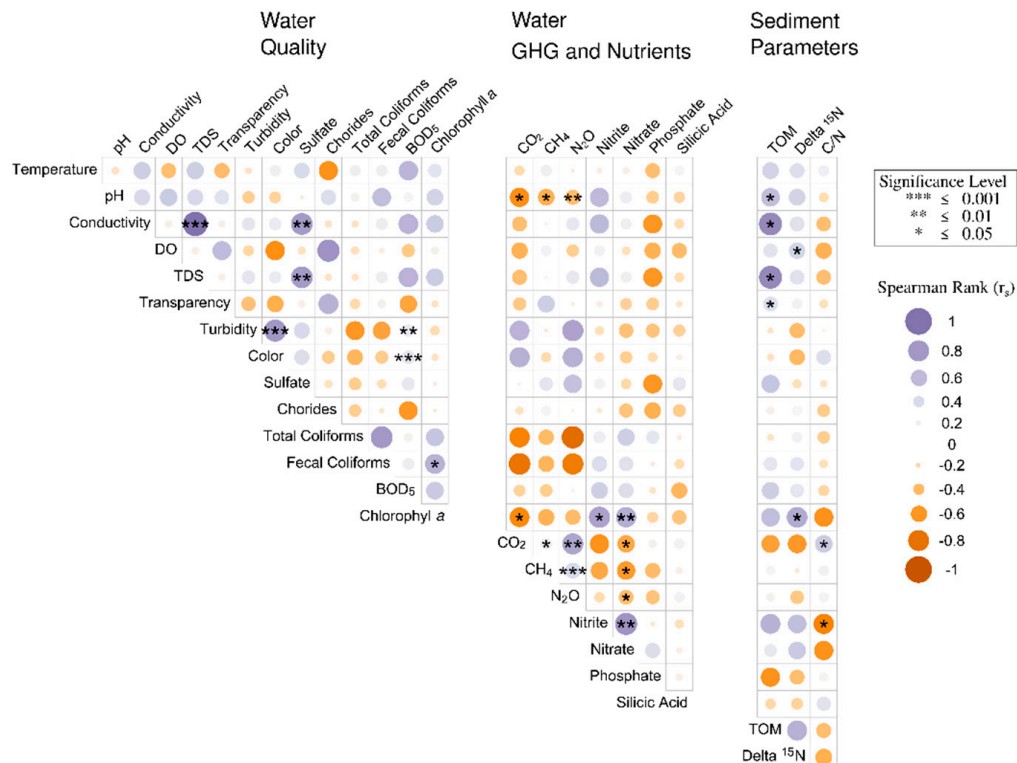

**Figure 6.** Spearman correlation analysis between water quality parameters, GHG, nutrients and sediment parameters.

### 3.5. Benthic Microbial Community Composition

The benthic microbial communities of El Sauce estuary were characterized by the presence of 56 bacterial and 8 archaea phyla (Supplementary Table S1). The microbial communities were dominated by Proteobacteria affiliated to Gammaproteobacteria class, mainly represented by the Betaproteobacterales order, followed by Bacteroidetes and Chloroflexi. Both spatial and temporal variations were observed, including changes in Chloroflexi that were predominantly detected towards the mouth of the estuary stations (E10 and E11), whereas Firmicutes and Verrucomicrobia had a higher relative contribution to the total microbial community towards the upper tributary (Stations F2 and F3), also showing slight changes in the winter versus the summer (Figure 7).

SIMPER analysis considering the spatial grouping, excluding Station E7, indicated that the core benthic microbial community, i.e., Proteobacteria, Bacteroidetes, and Verrucomicrobia, were more consistently present at the predischarge site compared to those in the discharge and post-discharge areas of the estuary (Figure S1). Discharge benthic microbial communities were differentiated by a greater contribution of Firmicutes and Spirochaetes and by the exclusive presence of Tenericutes, Lentisphaerae, Synergistetes, and LCP-89. In addition, Hydrogenedentes, Nitrospirae, Latescibacteria, Gemmatimonadetes, and Zixibacteria were exclusively found in the post-discharge area, whereas Actinobacteria, Acidobacteria, Planctomycetes, and Cloroflexi had a greater contribution (Figure S1).

In the pre-discharge area, Xhantomonadales, Sphingobacteriales, Cytophagales, Verrucomicrobiales, Flavobacteriales, and Pseudomonadales were among the twenty most abundant orders, being the most abundant ones during winter (Figure 8). In the discharge zone, Bacteroidales and Clostridiales were highly prevalent during winter and summer, whereas Desulfuromonadales, Syntrobacterales, Chitinophagales, Campylobacterales, and Spirochaetales had a greater contribution during summer. In the post-discharge zone, Desulfobacterales, Anaerolineales, and Myxococcales were abundant during winter and summer, while Steroidobacterales exhibited a greater contribution during summer (Figure 8).

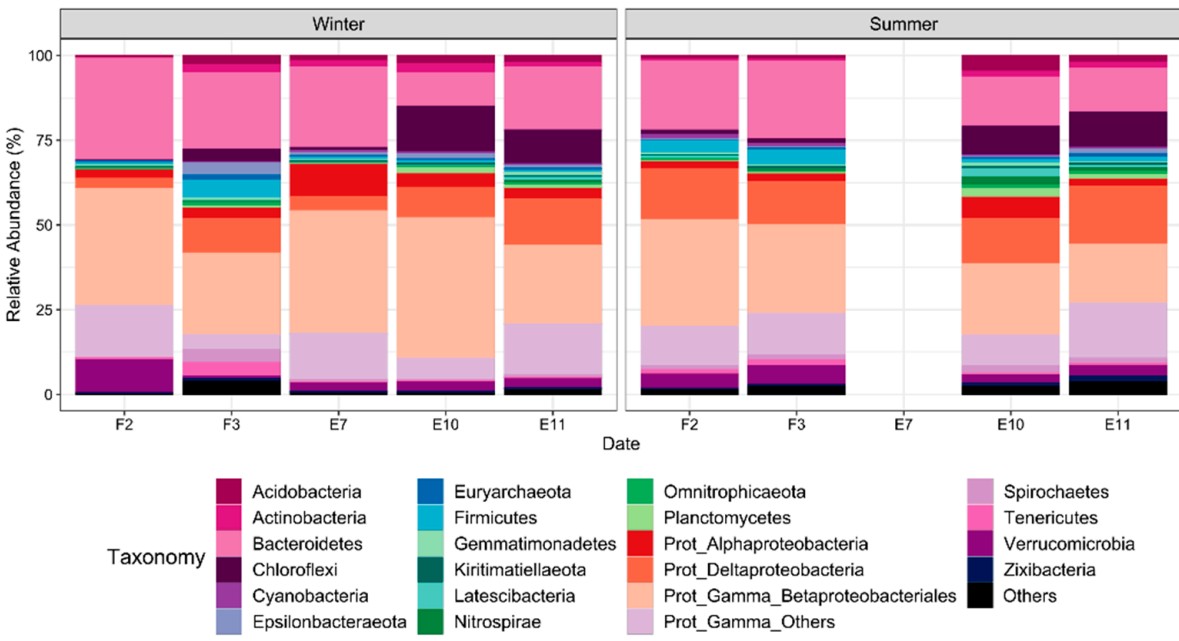

**Figure 7.** Taxa bar plot showed microbiome distribution from sediment at the phylum level during winter (left panel) and summer. Others correspond to phyla with relative abundance < %. Note that no sequencing data were available for E7 in summer.

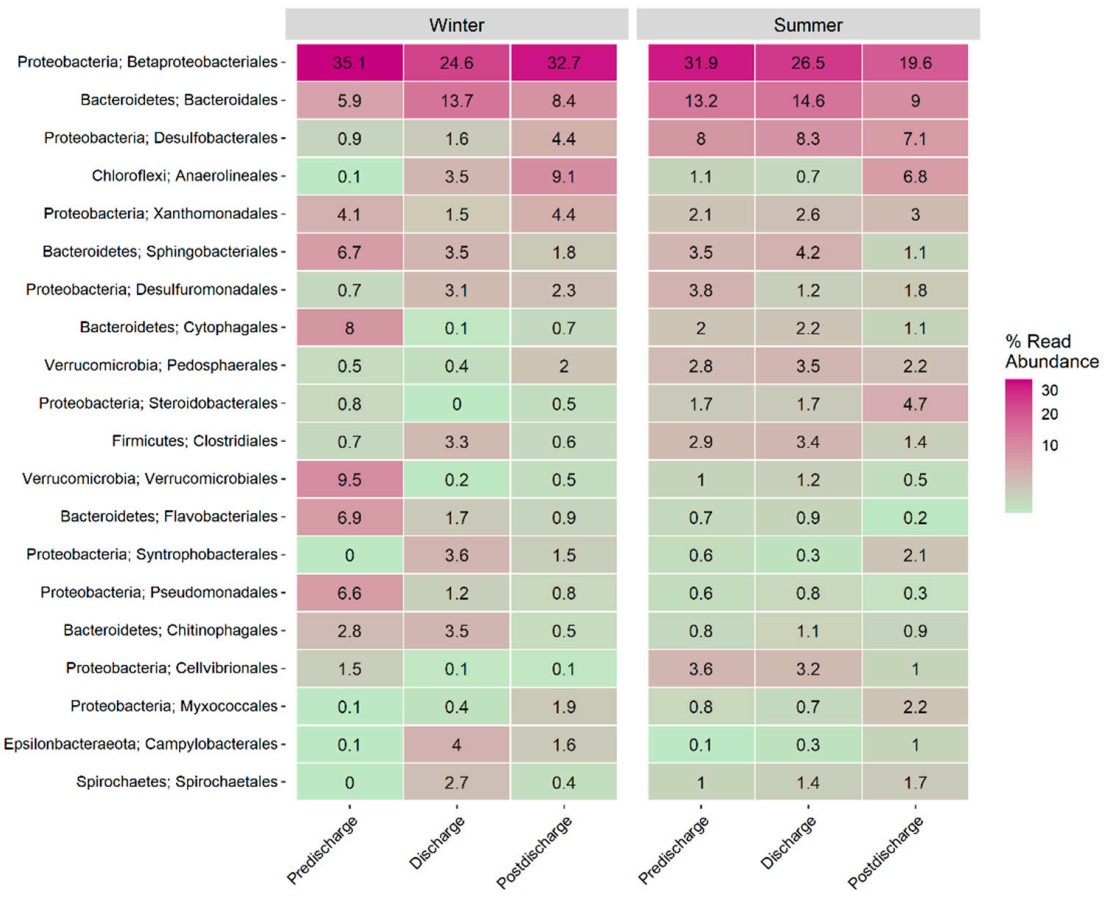

**Figure 8.** Heatmap showed the most dominant orders in the pre-discharge, discharge, and post-discharge stations during winter and summer.

Differential analysis based on the comparison of microbial communities at ASVs level showed a greater number of ASVs present in the discharge compared to those in the pre-discharge waters (Figure 9). Eighty-one ASVs predominantly affiliated to the orders Betaproteobacteriales, Bacteroidales, Syntrophobacterales, Campylobacterales, and Chitinophagales were more abundant ones in the discharge compared to those in the pre-discharge site, followed by a lower proportion of ASVs belonging to Spirochaetales, DTU014 (Clostridium), Izimaplasmatales, and Methanofastidiosales. In contrast, only eight ASVs affiliated with the orders Verrucomicrobiales and Flavobacteriales were enriched in the pre-discharge compared to those in the discharged water.

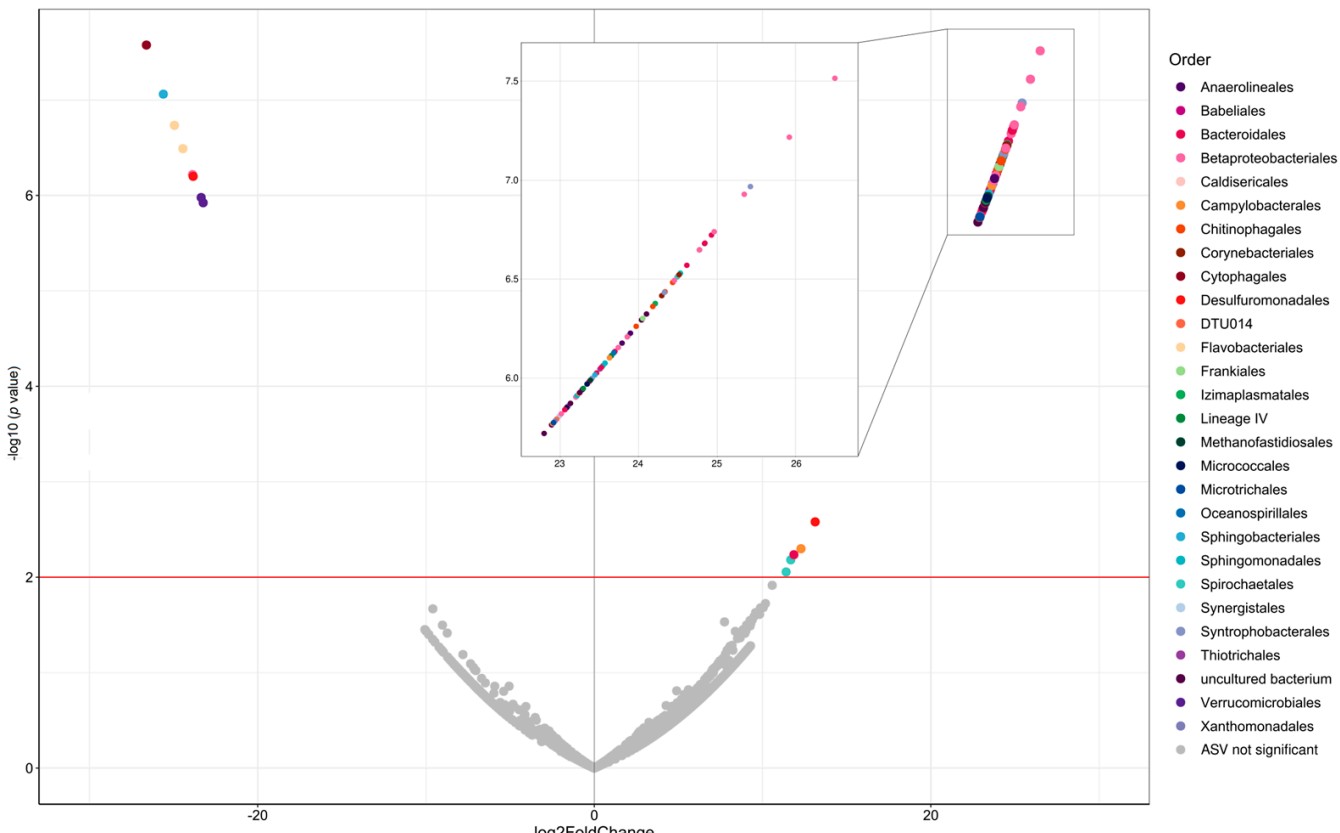

**Figure 9.** Volcano plot of differentially abundant ASVs identified between discharge and pre-discharge. A positive log2-fold change in relative abundance indicates the feature predominated in discharge water, and vice versa for pre-discharge water. The gray dots denote the ASVs without marked differences (*p*-value > 0.01, red line).

The redundancy analysis (RDA) confirmed the spatial variability of the microbial community structure at the phyla taxonomic level, revealing a clear separation between the discharge station (F3) and the mouth stations (E10 and E11) (Figure 10). Both the first and second axes account for 75.6% of the total variation in the benthic microbial community structure. Among the environmental variables accounting for the microbial community variability in the discharge waters, values of $N_2O$ and $CH_4$ and medium and coarse sands were significant, whereas the amount of very fine sand, TOM, pH, and $\delta^{15}N$ were significantly higher at the post-discharge stations E10 and E11 (Figure 10). Many water column quality parameters were associated with the benthic microbial community structure, but were not significant at a *p* value < 0.05.

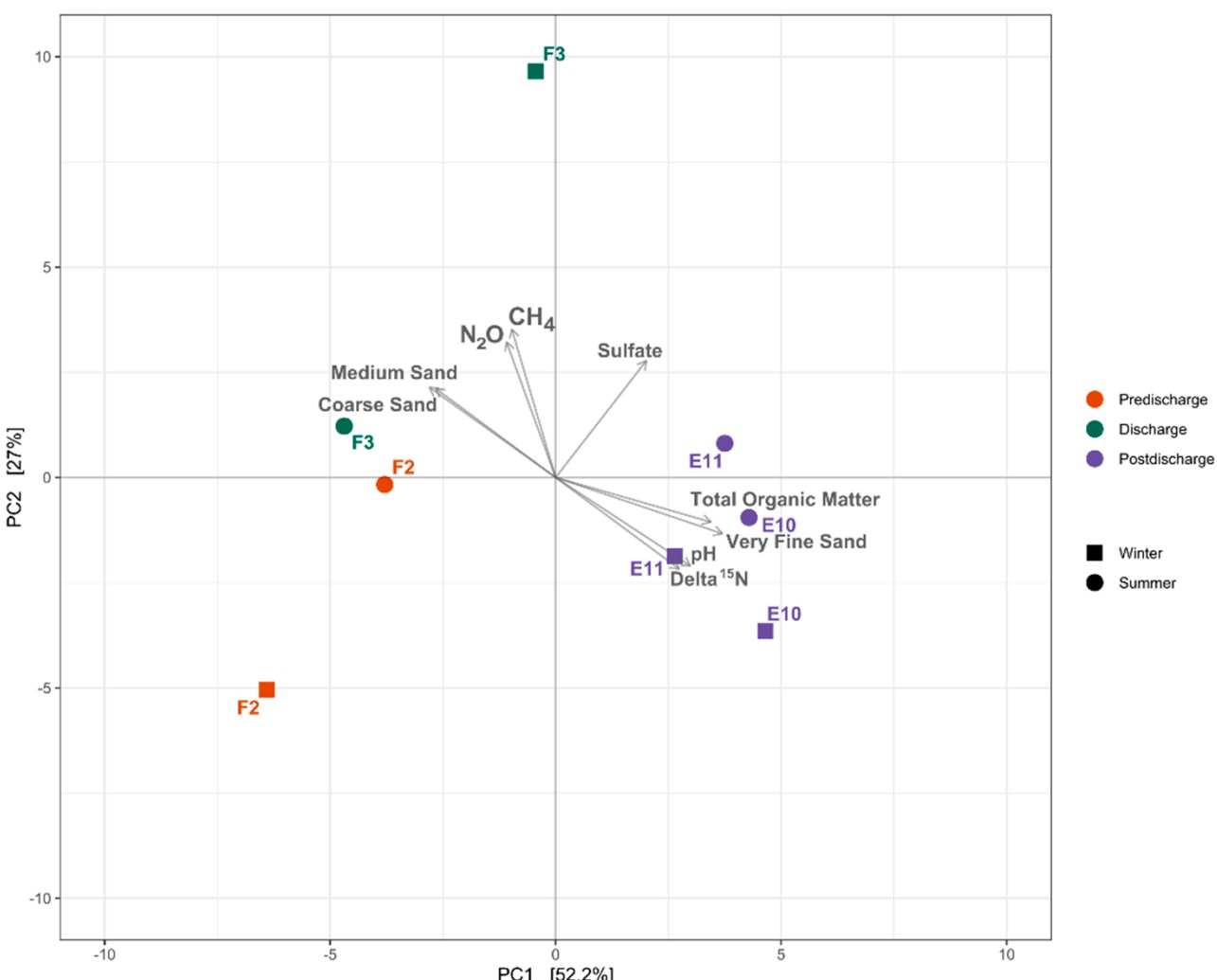

**Figure 10.** Bray–Curtis based redundancy analysis comparing benthic microbiomes dissimilarities between pre-discharge, discharge, and post-discharge stations and its relationship with physicochemical parameters. Parameters significantly correlated with the ordination identified through envfit function *p*-value < 0.05 were depicted in black.

*3.6. Functional Microbial Groups Spatial Variability during Winter and Summer in El Sauce Estuary*

The nitrifying communities' contribution based on 16S rRNA gene sequencing from the studied sediment presented distinct spatial patterns between winter and summer (Figure 11A). For instance, the Nitrosomonadaceae family was ubiquitous all along the stream during winter and summer, whereas Nitrospiraceae and ammonia-oxidizing archaea were detected mainly at the estuary mouth stations (Figure 11A). A higher richness of methanotrophs/methylotrophs were detected during winter compared to those in summer (Figure 11B), with the detection of Methylococcaceae at the discharge station. El Sauce benthic microbial communities were characterized by a rich methanogenic community that was particularly abundant at the discharge station (F3) in winter (Figure 11C). During winter, Methanobacteriaceae and Methanosarcinaceae were the predominant methanogens at the mouth stations (Figure 11C).

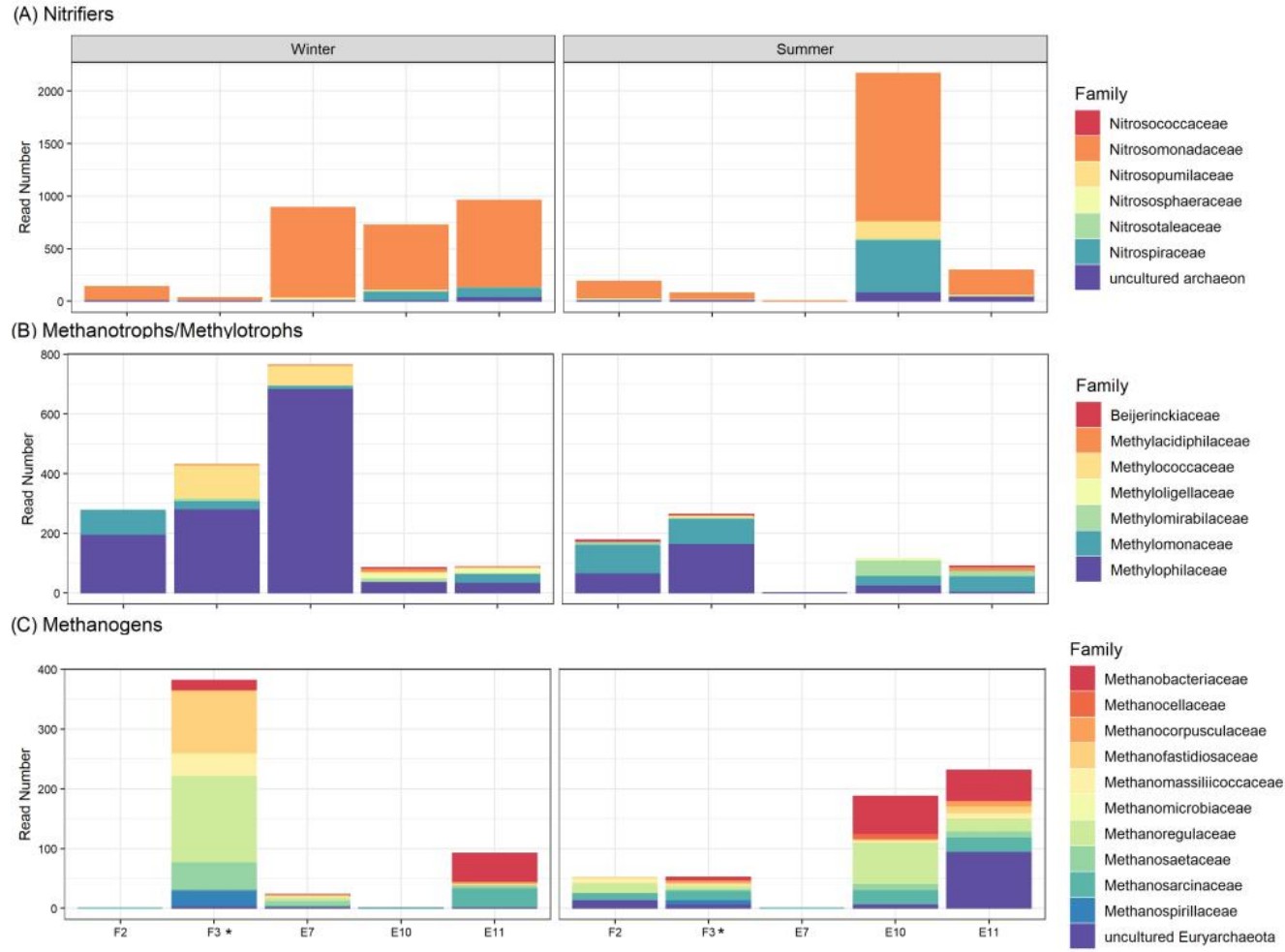

**Figure 11.** Taxa bar plot showed functional group 16S metabarcoding abundance at the family level (**A**) Nitrifyiers, (**B**) Methanotrophs/Methylotrophs and (**C**) Methanogens during winter and summer in all sampled sites. Note that no sequencing data were available for E7 in summer. * Discharge Station.

The abundance of the functional genes (*amoA, comaA, pmoA,* and *mcrA*) evaluated through qPCR showed a greater contribution of methanogens, ammonia, and methane oxidizers during summer compared to those in winter, especially at the estuary mouth (Figure 12A). In winter, the discharge station exhibited a lower abundance of ammonia and methane oxidizers compared to those of the pre- and post-discharge sites (Figure 12A–C). During summer, Betaproteobacteria ammonia oxidizers (βAOB) decreased in abundance towards the estuary mouth (Figure 12A), unlike comammox bacteria, which presented a similarly high contribution at all the sites, with a slightly greater contribution at the discharge station (Figure 12B). On the other hand, during winter and summer, methanogens abundance was higher in the discharge and post-discharge zones of the estuary (Figure 12D).

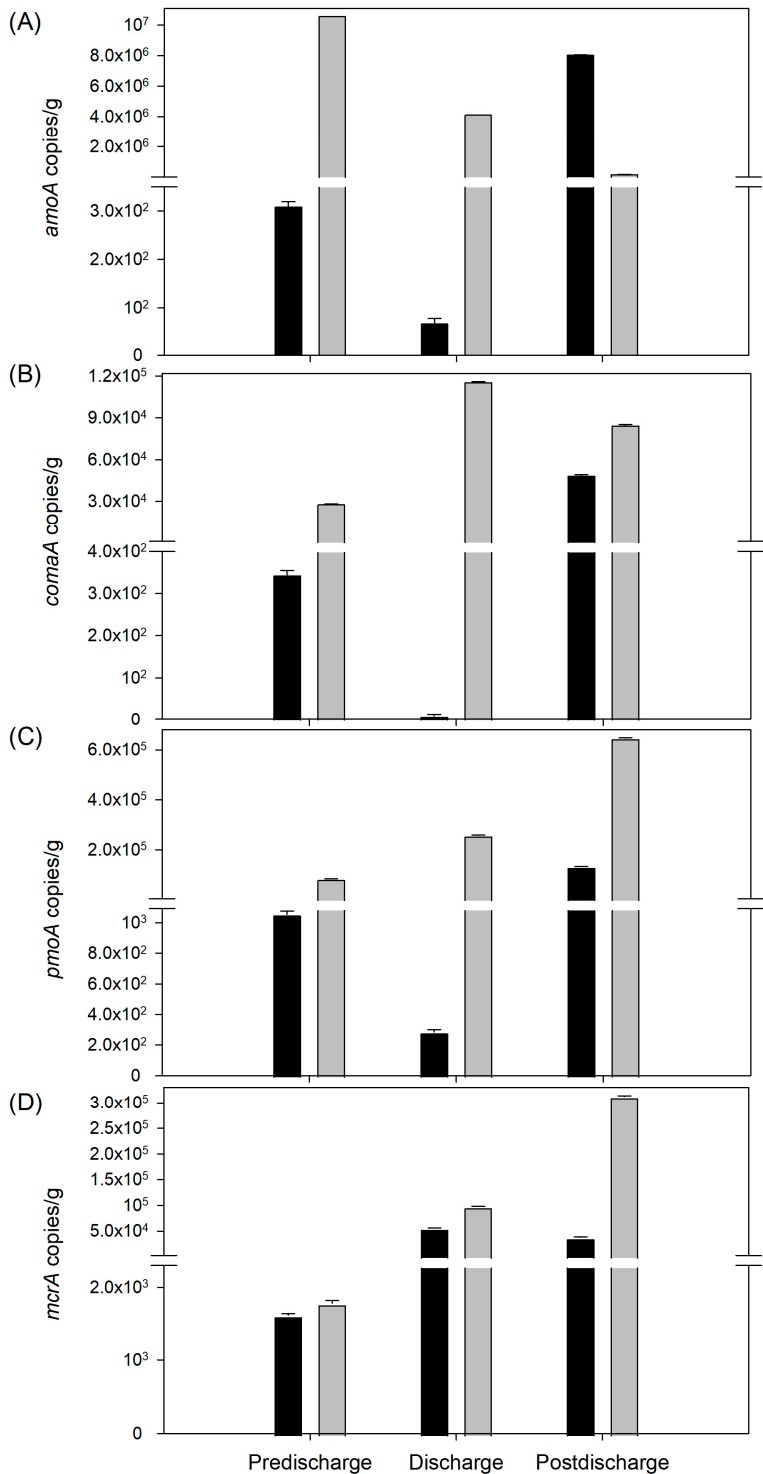

**Figure 12.** Abundance of all targeted functional genes (**A**) *amoA*, (**B**) *commaA*, (**C**) *pmoA*, and (**D**) *mrcA*) in the predischarge, discharge, and post-discharge stations during winter (black bars) and summer (grey bars). The abundance is expressed in copies per gram of sediment dry weight (DW).

Spearman multiple correlation analysis was carried out to evaluate the relationships between the functional groups abundance based on qPCR and the contribution of functional microbial groups composition by 16S rDNA sequencing. The *mcrA* gene abundance (i.e., in copies/g sediment DW) showed a positive correlation between several taxa of methanogens, including counts belonging to Methanomassilicoccaceae, Methanosaetaceae, Methanoregulaceae correlated with Methanospirillaceae, Methanofastidiosaceae, and the

total methanogens. On the other hand, methanotrophic communities quantification using *pmoA* gene copies/g sediment DW was positively correlated with the Methylacidiphilaceae family, whereas Methylococcaceae and Methylophilaceae families retrieved from 16S rRNA gene sequencing correlated positively with the total MOB (Figure 13). Neither βAOB nor commamox *amoA* gene copies/g sediment DW were correlated with the total nitrifiers communities recovered from 16S rRNA gene sequencing. However, counts belonging to several families of nitrifiers positively correlated with each other. This was the case for Nitrosopumilaceae and uncultured Crenarchaeota with Nitrosomonadaceae, Nitrosotaleaceae, and Nitropiraceae, whereas Nitrosomonadaceae and Nitrosotaleaceae were correlated with Nitropiraceae (Figure 13). Moreover, the functional microbial groups' abundance based on gene quantification qPCR were correlated with environmental factors (Table S2). Except for methanogens, all the functional groups were significantly correlated with $\delta^{15}N$ (*p*-value < 0.05). Additionally, comammox and methanotrophs were significantly correlated with temperature, $BOD_5$, and TOM (*p*-value < 0.05). The commamox was significantly correlated (*p*-value < 0.05) with conductivity and DTS, and methanogens were correlated with phosphate (Table S2).

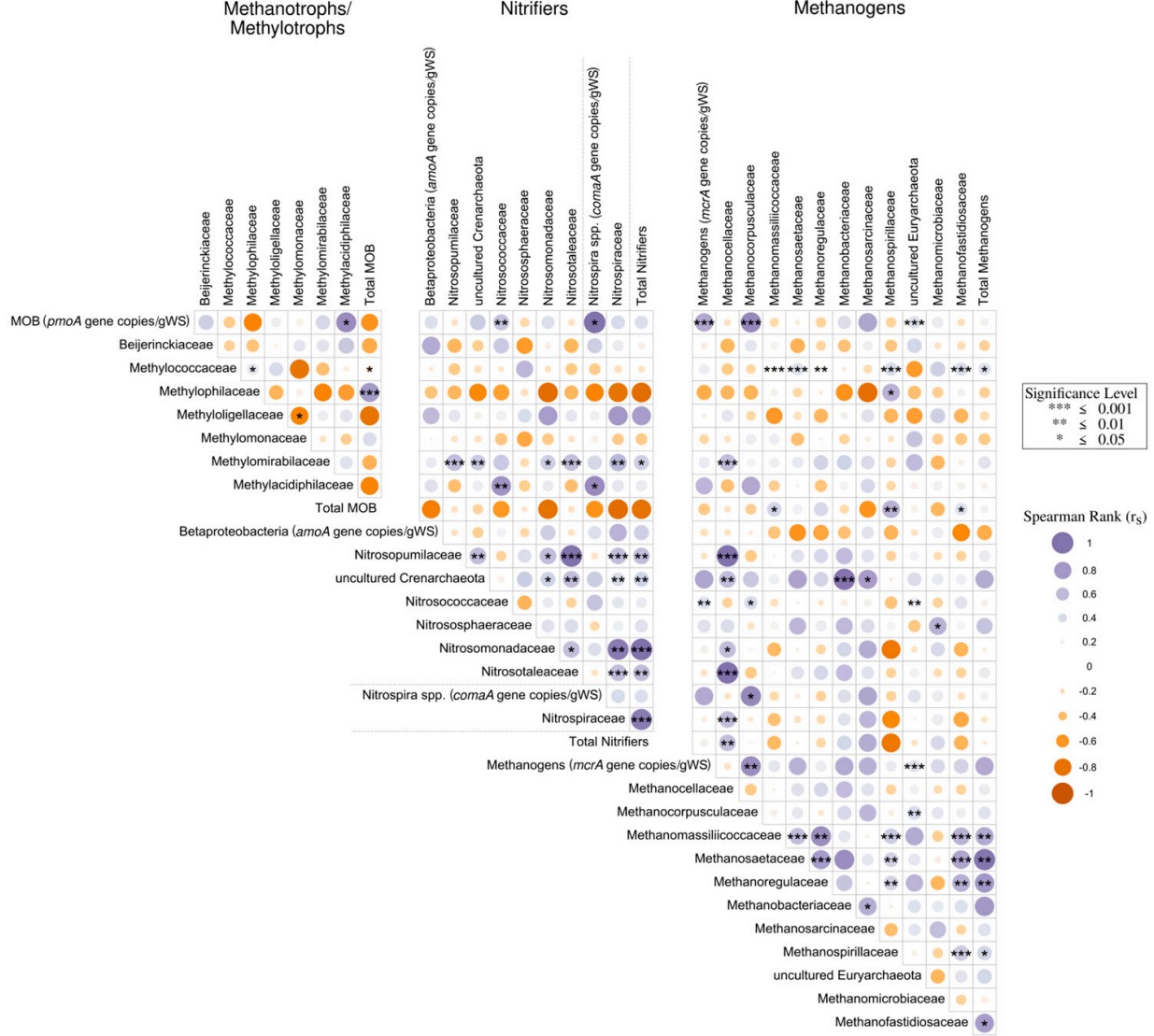

**Figure 13.** Spearman correlation analysis between functional microbial groups correlations based on qPCR genes abundances correlation with reads.

## 4. Discussion

El Sauce estuary is a fresh surface water body located in the Valparaíso region whose main water flow comes from Las Cenizas basin. Previous studies have reported that El Sauce is under anthropogenic pressure, categorizing the estuary as a permanent hyper-trophic estuarine environment mainly impacted by WWTP discharges, as well as percolated liquids from a nearby landfill and sewage discharges from houses not included in the sewage system (Rivera-Castro et al., 2020) [31]. These inputs are often composed of a mixture of partially degraded OM, nutrients, and microorganisms, many of them pathogens, and emerging contaminants that have the potential to reshape the natural community structure and biogeochemical activity, especially in semi-closed environments such as El Sauce. Our study reports the first evidence of the WWTP discharge's impact on the water quality, benthic microbial communities, and dissolved greenhouse gases of El Sauce estuary, providing new insights into how WWTP discharges influence the biogeochemistry of receiving water bodies, especially in complex and fragile environments such as the polluted coastal ecosystem of the Valparaiso region.

### 4.1. The WWTP Altered Physicochemical Properties of El Sauce Estuary

The physicochemical conditions of the watershed, such as color, transparency, turbidity, DTS, and total coliforms, suggest its poor water quality (Table 2). Moreover, the WWTP discharge (F3 station) considerably affected the physicochemistry of the estuary, decreasing the pH and the concentration of nitrate, nitrite, and phosphate compared to those levels in the upper tributary water in both winter and summer (Figure 2A,B). The DO decreased exclusively in summer, whereas the water temperature increased in both seasons. The discharge comes from Esval-Placilla wastewater treatment plant (WWTP), which includes an aerated lagoon and uses chloride gas bubbling as a secondary treatment, thus explaining the changes in the pH, temperature, and nutrient concentrations during our study. Our results agree with previous reports showing that El Sauce estuary is significantly affected by WWTP water discharge besides other sources of pollution, which have categorized the estuary as a eutrophic system for many years [31]. Moreover, in our study, the WWTP discharge station (F3) exhibited a twenty higher more dissolved $CH_4$ concentration compared to those of the other stations, including the stations situated at the mouth of El Sauce catchment area (Figure 3). Similar results have been found in sewage draining rivers and in the Guadalete estuary (e.g., Hu et al. 2018, Burgos et al. 2015) [61,62]. The $CH_4$ accumulation observed in the WWTP discharge station may be explained either by a stimulation of the methanogenesis or by the inhibition of $CH_4$ oxidation [63]. Additionally, the aerated lagoon may stimulate nitrification and denitrification processes, contributing to nitrogen loss by means of $NO$, $N_2O$, and $N_2$ production [64,65].

The isotopic composition of OM exhibited a marked spatial variability between the upstream stations (pre-discharge and discharge areas) and the mouth stations (post-discharge area) (Figure 5A). The values of the sedimentary isotopic signatures indicated that OM in the upstream station was associated with a terrestrial OM origin (from $\delta 13C = -28$ to $-29.5$ and from $\delta 15N = 4$ to 8.7), unlike those of the mouth of the estuary (from $\delta 13C = -25.3$ to $-26.5$ and from $\delta 15N = 8.4$ to 14.2) that evidenced a marine OM origin (Figure 4A). The OM isotopic $\delta 13C$ magnitudes were within values previously reported in estuaries in temperate areas, i.e., Godavari ($-24.6$), Loire ($-26.8$), Ems ($-24.2$), and the Sado ($-25.6$) mouth estuary [66,67]. Conversely, the OM isotopic $\delta 15N$ values determined in our study could be associated with various origins, including phytoplankton or residual anthropogenic nitrogen by rivers and diffuse runoff, i.e., $\delta 15N = 7.3 \pm 2.1$ % [68,69].

### 4.2. Benthic Microbial Communities and Functional Groups of El Sauce Estuary

El Sauce estuary holds a diverse benthic microbial community with the predominance of Proteobacteria, mainly Betaproteobacteriales and Bacteroidetes phyla (Figure 7). These results agree with microbial community composition reported from coastal wetlands of the semi-arid zone of central Chile [70] and with other tidal estuarine sediment

systems [71]. Regarding the water physicochemical properties, the benthic microbial community structure along El Sauce estuary was enriched with Firmicutes (Clostridiales order) and Bacteroidetes (Bacteroidales order), mainly in the WWTP discharge waters (Figure 7). These bacterial phyla are often dominant during wastewater treatment processes [72–74]. Moreover, in our study, Tenericutes, Lentisphaerae, Synergistetes, and LCP-89 phyla were exclusively detected in the WWTP discharge station (F3) and not in the other sampling areas (e.g., Figure S1). Moreover, a significant number of ASVs changed at the discharge station compared to those at the pre-discharge station (Figure 9). Firmicutes have been reported to be prevalent in the human stool microbiome, and therefore, are enriched in WWTP discharge [75,76]. In particular, the orders Clostridiales and Bacteroidales have been proposed as good bioindicators to estimate the degree of influence of sewage and fecal source in water catchment impacted by WWTP discharge [73,77,78]. However, its integration with alternative tools to identify and model the influence of WWTP discharge on the microbial communities in aquatic environments remains to be explored in depth.

In general, microbial groups belonging to Proteobacteria, Verrucomicrobia, and Actinobacteria were the dominant phyla in upstream waters of El Sauce estuary, as well as other wetlands of the semi-arid region of central Chile [70]. These groups were found to significantly decrease in the WWTP discharge station; however, they recovered again in the mouth of the estuary. Similar studies associated with WWTP discharge in urban and sub-urban rivers also found shifts in relevant phyla characterized by an increment in Verrucomicrobia and Actinobacteria and a decrease in Deltaproteobacteria [77].

Moreover, the phylum associated with functional groups, such as Cyanobacteria, showed a low abundance in all the sampled sites (Figure 7), which is probably attributed to the higher water turbidity, which decreases the solar radiation penetration to the sediment, thus inhibiting benthic photosynthetic processes [79]. In addition, other known functional taxa associated with nutrient recycling, such as the Nitrospirae and Chloroflexi phyla, were rare or undetected at the WWTP discharge station compared to those at the Estuary mouth (Figure 7). These phyla increased their relative sequence abundance towards the mouth (E10 and E11 stations) of the estuary (Figure 7), where higher dissolved nutrient concentrations were detected in the water. Nutrient availability and other compounds in the WWTP effluent could affect sensitive microbes such as Chloroflexi, as previously suggested [77].

The quantification of specific functional microbial groups by qPCR associated with nutrients and GHG recycling were influenced by the WWTP discharge and by the sampling season. The nitrifying ($\beta$AOB and comammox) and methanotrophic communities abundances decreased at the WWTP discharge station (F3) during winter (Figure 12A,C) and increased towards the mouth, which is important since they can still co-oxidize a variety of remaining xenobiotic compounds [80]. These results are in agreement with the decrease in the relative abundance of the Betaproteobacteriales order at the WWTP discharge station (Figure 9). Similarly, $\beta$AOB gene abundances were low in activated sludge from WWTP compared to those in coastal sediments ($5.68 \times 10^{-6} - 4.79 \times 10^{-5}$ versus $5.4 \times 10^{-1} - 34.4 \times 10^{-1}$ copies/g sample) [81] and between the sewage effluents in eutrophic lakes sediments, unlike in the adjacent areas [82]. Further research is required to elucidate if the decrease in Betaproteobacteriales is due to outcompetition with ammonia-oxidizing archaea and anaerobic ammonium-oxidizing bacteria or other specific environmental fluctuations.

In addition, all the targeted functional groups ($\beta$AOB, comammox, and methanotrophic) were enriched in the discharge area during summer, unlike winter, and in both seasons downstream at the estuary mouth (Figure 12A–C). In particular, a higher abundance of methanogenic communities was observed in the areas affected by the WWTP discharge (Figure 12D) than those in the other estuaries impacted by WWTP effluents [29]. Moreover, a rich methanogenic community was observed in winter at the discharge station and downstream based on 16S rRNA sequencing, accounting for the relevance of methanogenesis in wetlands.

The BOD$_5$ in the water, as well as the total OM and δ15N in the sediments, were significantly correlated with comammox and methanotrophic abundances. In general, both functional groups increased downstream towards the estuary mouth (Figure 12A–C), where greater δ15N values and more nutrients were found during both winter and summer (Figure 4A). Moreover, methanotrophic abundance in the estuary was significantly correlated with methanogens and with nitrifiers (Figure 13), indicating the potential metabolic interaction between these three functional groups. This is the case of nitrite-dependent anaerobic methane oxidation by bacteria, including Methylomirabilaceae (Figure 11), which could benefit from the nitrite produced by ammonia oxidation, linking carbon and nitrogen cycling in wetlands, as previously reported [83,84].

Benthic OM considering nitrogen availability could be relevant for the resistance or tolerance of the functional microbial groups to the WWTP discharge in the estuary. Winter conditions were unfavorable for methanotrophs and nitrifying assemblages, potentially reducing their role in nutrient and GHG recycling. Nutrient accumulation towards the mouth was evidenced during both seasons, confirming the pervasive eutrophic conditions of El Sauce estuary [31].

These results, together with previous studies on the same ecosystem, encourage further research on the current effects of WWTP discharges, an issue that has aroused wide public concern, especially in rural areas, since more infrastructure, resources, and higher regulatory standards are often oriented towards increasing the wastewater treatment capacity of cities [28,85]. This situation is even worse in countries such as Chile, where a relevant fraction of rural residents has a lower income, less education, and more importantly, less access to public utilities compared to those of their urban counterparts [86]. The effects on water availability and quality caused by these discharges and other sources of contamination are often suffered by more vulnerable rural communities that experience reduced fishery resources, agricultural production, and health [87].

This is the case of Laguna Verde and the community residing near El Sauce estuary, which is an area of rapid population growth in the Valparaiso area, and precarious urban development, and is highly dependent on agriculture [88]. Thereby, gaining insights into the interaction of the negative impacts that these discharges have on water quality is necessary to optimally address, not only their consequences on the environment at the local level, but also their effects on greenhouse gas dynamics.

## 5. Conclusions

In summary, our results confirm that El Sauce Estuary is an anthropogenically impacted ecosystem characterized by with poor-quality conditions in 40–60% of the variables. Among other anthropogenic pressures, including leachate from a municipal landfill and domestic water inputs that have permanently contributed to this eutrophic system, our study concluded that WWTP effluents play a major role in shaping the microbial and biogeochemical processes, resulting in GHG accumulation. For instance, WWTP effluent was found to have a greater impact in the upstream area compared to those in the pre- and post-discharge sites on the physicochemical conditions and the dissolved GHG concentration in the water, especially for CH$_4$ and N$_2$O by one order of magnitude. The organic matter quality varied spatially along the estuary, shifting upstream towards the mouth associated with terrestrial (δ13C = from −28 to −29.5 and δ15N = from 4 to 8.7, respectively) vs. estuarine (δ13C = from −25.3 to −26.5 and δ15N = from 8.4 to 14.2, respectively) origins. The benthic microbial community composition in the estuary followed the same trend as that of OM, characterized by the enrichment and enhancement of functional groups of methanotrophs and methanogens, reaching up to $6.5 \times 10^5$ and $2.1 \times 10^5$ abundance towards the mouth. However, the anthropogenic impact of the WWTP-discharge effluent upstream contributed to reshaping the benthic microbial community composition even at the phylum level by introducing allochthonous phyla and changing the relative contribution of core wetland taxa, thus potentially limiting the growth of nitrifying and methanotrophic bacteria during winter. Therefore, the spatio-temporal (winter vs. summer)

natural conditions and anthropogenic sources of perturbations strongly drive the dynamics of the benthic microbial community composition, and potentially, the way nutrients and GHG are recycled in El Sauce estuary.

**Supplementary Materials:** The following supporting information can be downloaded at: https://www.mdpi.com/article/10.3390/w15061251/s1, Figure S1: (A) Total organic matter percentage and (B) sediment granulometry distribution; Table S1: Benthic microbial community comparison between winter (W) and summer (S) sequences reads retrieved at El Sauce estuary sampling stations; Table S2: Spearman correlation analyses between functional group abundance quantification (qPCR) and the physicochemical variables studied.

**Author Contributions:** Conceptualization, F.P.-S., V.M. and R.O.; methodology, M.C.-D., C.L., C.A., C.R. and P.A.-M.; writing—original draft preparation, F.P.-S. and V.M.; writing—review and editing, F.P.-S., V.M., R.O., C.L., M.C.-D. and P.A.-M.; funding acquisition, V.M, M.C.-D. and R.O. All authors have read and agreed to the published version of the manuscript.

**Funding:** This research was funded by FONDECYT grant number 1211977 and Proyecto de Tesis de Postgrado at UPLA for FP and The APC was funded by the Dirección General de Investigación UPLA.

**Data Availability Statement:** Raw data supporting the conclusions of this article will be made available by the authors, without undue reservation.

**Acknowledgments:** We acknowledge N. Bassi and Mirian Lanfranco for their support in the field campaigns. We are grateful to Elias Arredondo, H. Peña, C. Sepúlveda, P. Reinoso, and A. Bello for their technical support in the laboratory for the molecular, nutrient, and GHG analyses. This research was technically supported by the GIIA Uplaguas, the supercomputing infrastructure of the National Laboratory for High Performance Computing, (ECM-02), FONDEQUIP EQM160131, and the Network Project RED21992 "Sistema articulado de investigación en Cambio Climático y Sustentabilidad en zonas costeras de Chile".

**Conflicts of Interest:** The authors declare no conflict of interest.

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
