# Peer review of "Microbial and Biogeochemical Shifts in a Highly Anthropogenically Impacted Estuary (“El Sauce” Valparaíso)"

_water, doi:10.3390/w15061251_

Round 1
Author Response
Enclosed is a revised version of Water paper no. water-2076788 entitled “Microbial and Biogeochemical Shifts in a Highly Anthropogenically Impacted Estuary (“El Sauce” Valparaíso)”. We thank the editor and two reviewers because their comments and questions have helped to improve the paper quality considerably.
One of the most relevant observations we received is related to the fact that the first version of the results and discussion sections of the manuscript have been predominantly focused on the effects of the WWTP discharge on the water quality of “El Sauce” estuary, overlooking other nonpoint source pollution, such as potential effect of upstream conditions or the influence of seawater intrusion from the sea. We agreed with the reviewer that the structure and organization of the discussion section could have induced the readers to make that direct link. Therefore, we made significant changes in the manuscript, including a deeper interpretation of available data, addition of several and more recent references and a complete reorganization of our discussion. We feel that far from being biased, the new version of the manuscript described a better picture of the microbial and biogeochemical shifts along the “El Sauce”.
Reviewer 1
1. This was an interesting study of the effects of WWTP and source water on water quality and microbial conditions along a freshwater and brackish estuary basin along the coast of Chile. The authors sampled above and below the point of discharge by the WWTP, and also near the estuary mouth. However, some elements of the manuscript were missing (such as a key figure) and the interpretations of some of the data seemed inconsistent to me. Especially concerning was the fact that the authors did not address the effect of upstream conditions, which appear to me to reflect considerable non-point source contamination. While the WWTP did seem to be a large contributor of two GHGs, it otherwise appeared to be releasing relatively high water quality from a nutrient standpoint. While it was the periodic source of potential pathogenic microbes, its releases appeared to be very similar to the contribution from the upstream source of water (the pre-discharge site). The study did not take into account marine contributions at all, which may have been considerable, given the nutrient profile at the mouth.
Response 1. We thank the reviewer for the comment. In fact, El Sauce is situated in an enclosed micro basin whose main natural water supply comes from limited water reservoirs Las Cenizas basin area, which is connected with “La Luz” reservoir, situated within the Peñuelas subwatershed representing a total of 320.9 km2 (https://www.scielo.org.mx/pdf/rica/v36n2/0188-4999-rica-36-02-261.pdf). In addition to that, it is important to note that a sandbar avoiding superficial water exchanges between the stream and the ocean is almost constantly present at the estuary and was present during both sampling periods. Previous report supports the low seawater influence at the mouth stations E10 and E11, located 892 and 127 meters away from the sea, respectively,(https://www.scielo.org.mx/pdf/rica/v36n2/0188-4999-rica-36-02-261.pdf) considering the subsaline conditions registered in the water.
In the new manuscript version we describe better the watershed conditions and sandbar presence in the 2.1 section “study area and sampling procedures” of Material and Method.
2. As a result, I was unconvinced of the paper’s main conclusion, that the WWTP was the main source of disruption to the water quality of the estuary. The authors’ appeared to be biased in their lack of serious discussion of other sources. While I agree that waste water should be held to high standards, it is unhelpful to focus on one contributor without engaging in the larger environmental issues that surround the ecosystem, especially when valuable resources may be allocated to one factor, neglecting others, and ultimately resulting in no substantial improvement to the ecosystem.
Response 2. We agree with the reviewer that the structure and style, especially of the beginning of the discussion and conclusion sections, seemed to be biased towards the effects of WWTP as the exclusive source of disturbance in the estuary. However, this happened because we clearly did not check carefully enough the style across the whole manuscript. The new version of the manuscript was checked and many editions were made in the discussion and conclusion to (i) uniform the style, (ii) add and highlight discussion related to biogeochemical indicators besides GHG that confirm a low quality of the water (BOD5,turbidity, color), and (iii) incorporate a deeper discussion about alternative sources of disturbance were also incorporated.
Line and section comments:
3. 27 “season” should be plural
Response 3: plural was added and grammatical issues were checked.
4. 72-74, 86 excessive use of commas
Response 4 : the use of commas was corrected and the English was checked.
5. Figure 1. The map is inadequate—too small and too low resolution. It is not helpful to show a blank inset map with political divisions. Better to show the region of Chile more clearly. The main map shows the main sample sites along the river, but doesn’t show the structure of the estuary mouth and the bay.
Section 2.1 What is the state of the estuary? Is it periodically open? How does water exchange when closed? What is the variation in flow across different seasons, including contributions (discharge) from the WWTP and from the river basin? Etc.
Response 5. A new map was created in order to show the region in Chile and the structure of the estuary mouth and the bay.
Thanks for the comment that allows us to explain with more details the estuary state and conditions. During both sampling a sandbar was present avoiding exchange with the ocean. This explanation as well as a better description of the watershed and the tributaries has been added in the 2.1 section “study area and sampling procedures”.
6. Table 1 is poorly formatted.
Response 6. The table format was fixed and followed the journal format requirements.
7. 193 “Data” is a plural word—use “Data were”
Response 7. Correction was made in the new version.
8. 237 show units.
Response 8. Thanks for the comment. Units were added in the new version of the manuscript.
9. 240-241 Statements like “the upper and intermediate pond tributary water” are not helpful to readers unfamiliar with the basin…which stations were these?
Response 9. We changed the statement for “Nutrients (i.e., nitrite, nitrate, silicate, and phosphate) at the F2 and F3 stations were characterized by extreme changes in nitrate and nitrite…”
10. 252 Opposite to what…unclear phrasing
Response 10. We changed the structure of the sentence to “Except for the upper tributary (Station F2) in summer, silicate concentrations presented lower concentrations during winter (320–100 µM) versus summer (100–250 µM) from F3 station towards the estuary mouth (Figure 2G and H)”
11. Fig 2 remove horizontal empty space between panels
Response 11. A new version of Figure 2 has been added.
12. Fig 3 is completely missing
Response 12: This figure was included after the editorial review, probably the reviewer checked the unrevised version, sorry for this mistake. Now we checked that all figures were embedded in the text.
13. Section 3.3 Figures of importance, like TOM, should be included in the main body.
Response 13: Thanks a lot for the comment. The figure Supplementary 1 became Figure 4 in the new version of the manuscript.
14. Fig 5. Indicate the meaning of the size and color of each circle. I struggle with the inclusion of sand particle size in this matrix or correlations. Any relationship will almost certainly correlative—like what relationship would exist between [N] and sand size? Nothing, because particle size is determined by water flow velocity, and not a mechanistic relationship. I would suggest simplifying the matrix to potential ecological/mechanistic relationships, which would be more informative to your story.
Response 14: We thank the reviewer for the comment. A new Fig 5, now Figure 6 was elaborated that incorporates size and color of circles as indicators of the Spearman Rank, to clarify the strength of the relationship between variables.
15. 329-331 “significantly present”. Significant at what level of confidence? I think you may just mean that they were consistently present.
Response 15. “significantly present” was changed for “consistently present” as the reviewer suggested.
16. Figs 7 and 8: I don’t see that Fig 7 provides any additional information than Fig 8…consider omitting 7.
Response 16. The water microbiome changes grouped as pre- discharge and post-discharged were reported at different taxonomic levels (phyla and order). Figure 7 of the original manuscript corresponds to a SIMPER analysis results at phylum level and Figure 8 to the heatmap associated with orders. We kept Figure 8 in the main text body and changed Figure 7 as a supplementary Figure 1S in the new manuscript version. Moreover we decided to add a new figure showing a higher taxonomic resolution in a volcano plot of differentially abundant ASVs identified between discharge and predischarge areas. We feel that this is the proper way to highlight ASVs that were predominantly detected at the pre-discharge vs. post-discharge stations. An explanation of this figure was added in the results section (3.5).
17. Fig 9 GHG are omitted, making it difficult to interpret the results, especially because other water quality parameters are not related to the discharge site.
Response 17. Figure was checked and GHG was enlarged for a better visualization. Water quality parameters are included in the analyses but were not shown in the previous figure version. Now we include them and edit the text to highlight their contribution.
18. 438-441 The authors are not candid about the contribution of the WWTP discharge: the water quality of the system is generally poor. At the discharge point, water quality improves, but temperature increases. The data suggest that non-point source pollution—or contributions from the Bay, which are never discussed-- is a greater problem than the WWTP for the estuary. The exceptions to this are the contributions, presumably from CH4 and N2O, but these are not included in the data presentation. It is unclear to me what the fate of these gasses are as they progress down the estuary: do they blow off, or do they continue to affect water quality downstream? I can’t say from the data presented.
Response 18. We agreed with the reviewer that the beginning of the discussion (423-436, first version) section could have seemed biased towards WWTP as the exclusive source of disturbance in the estuary. In consequence, the section was rewritten, but also, we highlighted previous study that refers to the WWTP as the main factor of perturbation of the water quality in “El Sauce”. Besides that, the water flow of the estuary is mainly dependent on rainfall and anthropogenic activities as previously reported by Rivera et al (2020). It is important to note that a sandbar avoiding superficial water exchanges between the stream and the ocean is almost constantly present at the estuary and was present during both sampling periods.
19. 475-485 This part of the Discussion continues to neglect the high levels of Firmicutes, Clostirdiales, and Bacteroides associated with F2 as well! Sometimes the rates are even higher than at F3. Removing potential pathogens should be a priority for the WWTP—but if there are non-point sources of pathogens from upstream, the impact will be small, especially given the recovery rates downstream. For me, the priority would be to connect more communities to a coherent system of waste/sewage collection, to minimize non-point contributions, which appear to have a pretty large effect on the system that is not discussed at all.
Response 19. The discussion was changed, including the subtitle to avoid bias associated with other potential non-point sources of human-derived bacteria. However, it is important to notice that discharge water presented a differential contribution compared to predischarge and postdischarge of several microbial taxa at different taxonomic levels, including phyla and ASVs which was highlighted.
20. 495 change “avoids” to ‘decreases solar penetration’
Response 20. Thanks for the comment. Correction was made in the text.
Reviewer 2 Report
This manuscript entitled “Microbial and Biogeochemical Shifts in a Highly Anthropogenically Impacted Estuary (“El Sauce” Valparaíso)“, seems logical work, and the authors achieved interesting results. However, there are several areas needs to be revised before acceptance. and there are some comments below to strengthen the work, may authors consider to get the work better. In general, I did not observe any fundamental and serious issue in this work, and I recommend publishing this work after considering a Major revision. In addition, authors must have to send their manuscript for English Editing to remove grammar and sentence mistakes.
*Based on MS content, it’s better to alter title to fit the content and clearly represent the paper topic.
*Abstract, authors must define the characterization technique names when write very first time instead of acronym.
*The novelty of the work has to be highlighted in the abstract.
*Keywords should reflect the key points of the work. Be corrected.
*Authors must modify introduction by stating the following points "Problems, Possible solution, Disadvantages of these solutions, Author's idea, Advantages of authors idea, etc." Introduction in overview form not recommended. Please follow the following references for the modification of the introduction and must cite them.
https://doi.org/10.1016/j.envres.2022.114270https://doi.org/10.1016/j.envres.2022.114113
https://xs2.zidianzhan.net/scholar?oi=bibs&hl=zh-CN&cluster=7479047946711920426
https://doi.org/10.1007/s11244-021-01498-x
*Please explain about the significance, novelty, and importance of current work and it should be compared to previously literature that published at this filed.
*In experimental Section, the purity of each used chemicals compound must be provided by authors.
*Each characterization method must be provided with proper citation.
*The quality of Figs. 1&5&6&12 must be enhanced.
*The format of symbol and unit should be double-checked.
*A comparative for the result of the present work with the previous works should be presented as Table, it could show the superiority of the present work.
*The numbering of paragraphs has some mistakes, check and fix them.
*Authors need to modify conclusion with more data.
Author Response
Enclosed is a revised version of Water paper no. water-2076788 entitled “Microbial and Biogeochemical Shifts in a Highly Anthropogenically Impacted Estuary (“El Sauce” Valparaíso)”. We thank the editor and two reviewers because their comments and questions have helped to improve the paper quality considerably.
One of the most relevant observations we received is related to the fact that the first version of the results and discussion sections of the manuscript have been predominantly focused on the effects of the WWTP discharge on the water quality of “El Sauce” estuary, overlooking other nonpoint source pollution, such as potential effect of upstream conditions or the influence of seawater intrusion from the sea. We agreed with the reviewer that the structure and organization of the discussion section could have induced the readers to make that direct link. Therefore, we made significant changes in the manuscript, including a deeper interpretation of available data, addition of several and more recent references and a complete reorganization of our discussion. We feel that far from being biased, the new version of the manuscript described a better picture of the microbial and biogeochemical shifts along the “El Sauce”.
Reviewer 2
This manuscript entitled “Microbial and Biogeochemical Shifts in a Highly Anthropogenically Impacted Estuary (“El Sauce” Valparaíso)“, seems logical work, and the authors achieved interesting results. However, there are several areas needs to be revised before acceptance. and there are some comments below to strengthen the work, may authors consider to get the work better. In general, I did not observe any fundamental and serious issue in this work, and I recommend publishing this work after considering a Major revision. In addition, authors must have to send their manuscript for English Editing to remove grammar and sentence mistakes.
1.*Based on MS content, it’s better to alter title to fit the content and clearly represent the paper topic.
Response 1. Thanks for the comment, however, we think that the manuscript title does reflect the content of the study which is to characterize the benthic microbial composition and biogeochemical conditions in a long term anthropogenic impacted estuary. However, some sections of the manuscript were refocused to avoid confusion, especially in the discussion and conclusion as requested.
2.*Abstract, authors must define the characterization technique names when write very first time instead of acronym.
Response 2. Definition of specific names were included in the abstract and acronyms were given after their corresponding first mention.
3.*The novelty of the work has to be highlighted in the abstract.
Response 3. We add a paragraph to highlight the novelty of the work which is a multidisciplinary approach. Besides describing only water quality parameters, our study deals with molecular characterization of microbial communities and biogeochemical variables determined via sedimentary conditions (isotopic signatures) and greenhouse gases distribution.
4.Keywords should reflect the key points of the work. Be corrected.
Response 4. Keywords were changed to the following (wetland; greenhouse gases; organic matter; benthic microbial community; nitrifiers; methanotrophs; methanogens)
5.*Authors must modify introduction by stating the following points "Problems, Possible solution, Disadvantages of these solutions, Author's idea, Advantages of authors idea, etc." Introduction in overview form not recommended. Please follow the following references for the modification of the introduction and must cite them.
Response 5. Thanks for the comment and listed references. By reviewing the content of the reference list, we found out that they are not adequate/accurate to our manuscript scope that aimed to characterize the microbial community and biogeochemical conditions in an estuary. Therefore, the study is not focused on solutions or innovation associated with water treatment as can be seen in the given references list.
https://doi.org/10.1016/j.ceramint.2021.08.042 “Antibacterial and photocatalytic behaviour of green synthesis of Zn0.95Ag0.05O nanoparticles using herbal medicine extract”
https://doi.org/10.1016/j.envres.2022.114270 “Synthesis and characterization of g–C3N4–CoFe2O4–ZnO magnetic nanocomposites for enhancing photocatalytic activity with visible light for degradation of penicillin G antibiotic”
https://doi.org/10.1016/j.envres.2022.114113 “Recent progress on adsorption of cadmium ions from water systems using metal-organic frameworks (MOFs) as an efficient class of porous materials”
https://xs2.zidianzhan.net/scholar?oi=bibs&hl=zh-CN&cluster=7479047946711920426 “Determination of tert-butylhydroquinone using a nanostructured sensor based on CdO/SWCNTs and ionic liquid”
https://doi.org/10.1007/s11244-021-01498-x “A Tramadol Drug Electrochemical Sensor Amplified by Biosynthesized Au Nanoparticle Using Mentha aquatic Extract and Ionic Liquid”
https://doi.org/10.1016/j.jwpe.2022.102696 “Magnetic-MXene-based nanocomposites for water and wastewater treatment: A review”
https://doi.org/10.1016/j.chemosphere.2022.134595 “Lotus seedpods biochar decorated molybdenum disulfide for portable, flexible, outdoor and inexpensive sensing of hyperin”
6.*Please explain about the significance, novelty, and importance of current work and it should be compared to previously literature that published at this filed.
Response 6. The significance of the study was given in the introduction and paraphrased in the paragraph 93 - 104. Also, the discussion section was rewritten in order to highlight the novelty and the significance of the current study between the lines 624 - 639.
7.*In experimental Section, the purity of each used chemicals compound must be provided by authors.
Response 7. A general paragraph was included in the new manuscript version since all the chemicals used are “analytical grade” as suggested by the references.
8.*Each characterization method must be provided with proper citation.
Response 8. Citation was provided for all the methods used for analytical procedures.
9.*The quality of Figs. 1&5&6&12 must be enhanced.
Response 9. The quality of figures 1, 5, 6 and 12 were edited and improved.
10.*The format of symbol and unit should be double-checked.
Response 10. All symbols and units were checked by more than one author of the manuscript.
11.*A comparative for the result of the present work with the previous works should be presented as Table, it could show the superiority of the present work.
Response 11. The only parameters previously reported in the El Sauce Estuary correspond to water quality values. In the new manuscript version this data was included in Table 2 for comparison purposes.
12.*The numbering of paragraphs has some mistakes, check and fix them.
Response 32. The numbering of paragraphs was corrected according to the reviewer´s instructions.
13.*Authors need to modify conclusion with more data.
Response 13. Thanks for the comment, relevant data were included in the conclusion to highlight the main results of our study.
Reviewer 3 Report
The manuscript is well written and the topic is interesting.
Introduction is enough and gives general idea. Material and methods are well explained. Results are well presented.
Discussion part is including some results. Remove that parts to results or change title as results and discussion (ıf possible for journal rules).
The conclusion can be improved. Instead of summary of the study important findings of the study can be stated (please see the attachment).
Please see the attachment for other comments and edits.
The manuscript can be accepted for publication after minor revision.

Author Response
Reviewer 3
- The manuscript is well written and the topic is interesting.
Introduction is enough and gives general idea. Material and methods are well explained. Results are well presented.
R1. We thanked the reviewer for the comments.
- Discussion part is including some results. Remove that parts to results or change title as results and discussion (ıf possible for journal rules).
R2. We thank the reviewer for the suggestions, however, we respectfully declined the suggestion. We based that decision on two elements. The first one is that there are no new results on the Discussion section (that had not been mentioned in the results section). The second reason is that in the first round of revisions we were asked to devote enough space on the effects of the WWTP discharge as well as other nonpoint source pollution, such as potential effect of upstream conditions or the influence of seawater intrusion from the sea on the water quality of “El Sauce” estuary. Hence, there is the need to develop such arguments, in a new section as Discussion, as all data pieces can be analyzed together.
- The conclusion can be improved. Instead of summary of the study important findings of the study can be stated (please see the attachment).
- Please see the attachment for other comments and edits. The manuscript can be accepted for publication after minor revision.
- Line 66. Add relevant references
- References were added.
- Line 110. Change field campaigns for field work.
- Change was made.
- Line 185. Remove space
- Space was removed.
- Line 189. what about E7?
- Sentence “Samples from station E7 were not included in microbial communities analysis.” was added.
- Line 269. Using only transparency can be misleading. it may be better to use Secchi (transparency/max. depth better
- Secchi disk was used as part of the standard methods for transparency, we add this in the table to clarify
- Fix figure 4.
- Figure 4 was fixed.
- it may be good idea to add this figure as main figure
- We kept Figure S1 (that was former figure 7 of the first version of the manuscript) as supplementary, since we were asked by another reviewer to move to Supplementary.
- Lines 466 and 578. importance of the study remove to the conclusion part.
- We gently refused the recommendation of both sentences (466 and 578) because a very important part of reviewing discussion was based on the argument that the manuscript was predominantly focused on the effects of the WWTP discharge on the water quality of “El Sauce” estuary, overlooking other nonpoint source pollution. Therefore, in the second version of the manuscript we reorganized the discussion and conclusion sections, avoiding such interpretations.
- Line 523. add references.
- References were added. We deleted by mistake.
- Line 614. make a space down the title
- Space was made.
- Line 624. Delete text.
- Text was deleted.
- Line 630. Delete text.
- Text was deleted.
Round 2
Reviewer 2 Report
Accept
Author Response
We thank reviewer 2 for the comments, we significantly improved the manuscript thanks to three reviewers suggestions. English style and figures were also checked in the second round.